# BOOSTED CURRICULUM REINFORCEMENT LEARNING

**Pascal Klink,** **Carlo D'Eramo, Jan Peters**
Department of Computer Science
TU Darmstadt, Germany

**Joni Pajarinen**
Department of Electrical Engineering
and Automation
Aalto University, Finland

## ABSTRACT

Curriculum value-based reinforcement learning (RL) solves a complex target task by reusing action-values across a tailored sequence of related tasks of increasing difficulty. However, finding an exact way of reusing action-values in this setting is still a poorly understood problem. In this paper, we introduce the concept of boosting to curriculum value-based RL, by approximating the action-value function as a sum of residuals trained on each task. This approach, which we refer to as boosted curriculum reinforcement learning (BCRL), has the benefit of naturally increasing the representativeness of the functional space by adding a new residual each time a new task is presented. This procedure allows reusing previous action-values while promoting expressiveness of the action-value function. We theoretically study BCRL as an approximate value iteration algorithm, discussing advantages over regular curriculum RL in terms of approximation accuracy and convergence to the optimal action-value function. Finally, we provide detailed empirical evidence of the benefits of BCRL in problems requiring curricula for accurate action-value estimation and targeted exploration.

## 1 INTRODUCTION

The combination of reinforcement learning (RL) (Sutton & Barto, 2018) algorithms with powerful function approximators, i.e., deep RL (François-Lavet et al., 2018; Mnih et al., 2015), is a breakthrough towards solving complex decision making and control problems that were impractical for previous shallow RL methods. However, the outstanding performance of deep RL techniques is obtained at the cost of a massive amount of data needed from the interaction with the environment, hindering the practicality of deep RL methods.

Curriculum RL is a biologically inspired approach that frames the problem of learning a complex target task into learning a sequence of simplified, increasingly difficult, versions of it (Florensa et al., 2017; Shao et al., 2018; Ivanovic et al., 2019; Narvekar et al., 2020). Commonly, establishing the complexity of a task in RL is not trivial, as an indisputable notion of complexity is still missing. However, from a practical point of view, we can consider a task complex if it requires targeted exploration in the environment or complex policies to be solved. In curriculum RL, the notion of task complexity is key, and it is assumed that tailored tasks in a sequence of increasing complexity are presented to the learning agent. Such an appropriate design is, however, a difficult problem, requiring either human expertise (Narvekar et al., 2020) or solutions to automatize the selection of tasks following the learning of the agent (Jiang et al., 2015; Svetlik et al., 2017; Klink et al., 2020). Regardless of using a handcrafted or an automatically generated curriculum, a common approach adopted by most value-based curriculum RL methods is to use the same function approximator throughout all the tasks (Narvekar et al., 2020): given a task, an approximation of the action-value function is learned, and used as an initialization for learning the action-value function of the next task in the curriculum. For the success of this procedure, the function approximator needs to be powerful enough to handle the complexity of the target task. However, this requirement raises the key issue of adequately designing the function approximator. A complex function approximator is prone to overfitting, and sensitive to hyper-parameters, being detrimental for learning simple tasks in the curriculum; conversely, using an overly simple function approximator can hinder the learning of the target task. Moreover, choosing an adequate function approximator *a priori* is not trivial, as the target task is only visited at the end, and it is commonly unsolvable by non-curriculum RL methods.

---

*Correspondence to Pascal Klink: `pascal@robot-learning.de`.

In this paper, we introduce a novel approach that models the action-value function of a task as a sum of residuals learned on the previous tasks in the curriculum. Our method increases the representativeness of the function approximator as new tasks are presented to the agent, leading to a procedure that increases the size of the functional space as the curriculum proceeds. This results in a learning procedure that increases the power of the function approximator according to the complexity of the task at hand, using small models in simple problems to avoid overfitting, and using large models (resulting from the sum of small ones) to handle the dimensionality of complex problems. Notably, if the curriculum is sufficiently fine-grained, the residuals become smoother functions, close to zero, that are easy to learn, even if the respective tasks are harder than the previous ones. We call our method boosted curriculum reinforcement learning (BCRL), for its resemblance with the boosting technique in supervised learning (Freund, 1995). We provide a theoretical analysis of our approach under the lens of the approximate value iteration (AVI) framework (Farahmand, 2011). In AVI, an estimate of the optimal action-value function is obtained as an iterative process that starts from an arbitrary estimate and applies the optimal Bellman operator until convergence. It is shown that two sources of error exist in AVI: (i) computation of the optimal Bellman operator, (ii) representation of the action-value function (Farahmand, 2011). While the former is due to the need of using samples to approximate the unknown optimal Bellman operator, the latter depends on the representativeness of the functional space chosen to approximate the action-value function. In this work, we formalize the curriculum RL problem in the AVI setting to investigate the representation of the action-value function obtained by BCRL and cases in which this representation may result in a tighter bound on the approximation error compared to regular curriculum RL. We complement this analysis with an empirical evaluation of BCRL in AVI and RL settings, resorting to the fitted $Q$-iteration algorithm (Ernst et al., 2005), least-squares policy iteration (Lagoudakis & Parr, 2003), and deep $Q$-networks (Mnih et al., 2015), demonstrating advantages of BCRL over regular curriculum RL.

## 2  RELATED WORK

Learning under a dynamic set of training experience has been investigated in different fields throughout decades (Saul, 1941; Elman, 1993; Asada et al., 1996; Krueger & Dayan, 2009). A recent paper by Bengio et al. (2009) has established the term *curriculum learning* for the concept of organizing training data of a neural network in a beneficial way for performance and generalization. This term carried over to the domain of RL, where curricula over training tasks have been shown to stabilize the highly challenging optimization problem of RL (Narvekar et al., 2020). While a large body of research in the domain of curriculum reinforcement learning investigates the interesting problem of optimally selecting and scheduling the training tasks for an RL agent to optimize performance on a set of target tasks (Florensa et al., 2017; Ivanovic et al., 2019; Shao et al., 2018; Klink et al., 2020), the architecture of the learning agent is often assumed fixed and the transfer of agent behavior between subsequent tasks in the curriculum is done by this shared fixed architecture. However, aside from investigations into the benefit of an adequately chosen evolution of training experience, investigations of an optimal learning architecture (Elsken et al., 2019) and its evolution during learning (Aoki & Siekevitz, 1988; Rumelhart et al., 1988; Ash, 1989; Fahlman, 1990; Lahnajärvi et al., 2002; Nitanda & Suzuki, 2018) have been conducted in supervised learning scenarios. The adaptation of these methods to the setting of RL has been limited to on-policy temporal-difference learning (Rivest & Precup, 2003; Vamplew & Ollington, 2005b;a) and it has not been further pursued in recent years. We shortly review further approaches to knowledge transfer in Appendix A.

Our work aims to improve the information transfer between subsequent tasks in curriculum value-based reinforcement learning via a boosting procedure (Freund, 1995). The benefit of the boosting framework is that it is well understood from a theoretical perspective (Tosatto et al., 2017), allowing for the derivation of error bounds, as also presented in this paper. Further, it is agnostic to any value-based RL algorithm, avoiding the need to adjust the RL algorithm of choice to be suited for the evolution of the function approximator (Rivest & Precup, 2003; Vamplew & Ollington, 2005a;b).

## 3  PRELIMINARIES

### 3.1  DEFINITIONS AND NOTATION

We recall the notions about Markov Decision Processes (MDPs) (Puterman, 1990), and the notation used for approximate value iteration (AVI) (Farahmand, 2011), that are relevant to our work.

For a space $\Sigma$, we define $\mathcal{M}(\Sigma)$ as the set of probability measures over the $\sigma$-algebra $\sigma_\Sigma$. We denote $\mathcal{B}(\Sigma, B)$ as the space of bounded measurable functions w.r.t. $\sigma_\Sigma$ with bound $0 < B < \infty$. A finite-action discounted MDP is defined as a tuple $\mathcal{T} = \langle \mathcal{S}, \mathcal{A}, \mathcal{P}, \mathcal{R}, \gamma \rangle$, where $\mathcal{S}$ is a measurable state space, $\mathcal{A}$ is a finite set of actions, $\mathcal{P} : \mathcal{S} \times \mathcal{A} \to \mathcal{M}(\mathcal{S})$ is a transition kernel, $\mathcal{R} : \mathcal{S} \times \mathcal{A} \times \mathcal{S} \to \mathbb{R}$ is a reward function, and $\gamma \in [0, 1)$ is a discount factor. A policy $\pi$ is a mapping from the state space $\mathcal{S}$ to a probability distribution over the action space $\mathcal{A}$. A policy induces an action-value function $Q^\pi(s, a) \triangleq \mathbb{E}\left[\Sigma_{t=0}^\infty \gamma^t R_t | S_0 = s, A_0 = a\right]$, corresponding to the expected cumulative discounted reward obtained performing action $a$ in state $s$ and following the policy $\pi$ thereafter. RL aims at solving an MDP by finding an optimal policy $\pi^*$ which induces an optimal action-value function $Q^*(s, a) = \sup_\pi Q^\pi(s, a)$. For immediate rewards uniformly bounded by $R_{\max}$, the action-value function is bounded by $Q_{\max} = R_{\max}/1-\gamma$. For a probability measure $\mu \in \mathcal{M}(\mathcal{S} \times \mathcal{A})$, we define the $L_p(\mu)$-norm of a measurable function $Q \in \mathcal{B}(\mathcal{S} \times \mathcal{A})$ as $\|Q\|_{p,\mu} \triangleq \left[\int_{\mathcal{S} \times \mathcal{A}} |Q(s, a)|^p d\mu(s, a)\right]^{1/p}$. Given a sequence of values $x_{1:n} = (x_1, \ldots, x_n)$ for some space $\mathcal{X}$, the empirical $L_p$-norm of a function $f : \mathcal{X} \to \mathbb{R}$ is $\|f\|_{p,x_{1:n}} \triangleq \left[\frac{1}{n} \sum_{i=1}^n |f(x_i)|^p\right]^{1/p}$. Conveniently, when $X_i \sim \mu$, we have $\mathbb{E}\left[\|f\|_{p,x_{1:n}}\right] = \|f\|_{p,\mu}$. From now on, we consider the $L_2$-norm when the subscript $p$ is omitted.

**Operators.** We define the optimal Bellman operator $T^* : \mathcal{B}(\mathcal{S} \times \mathcal{A}) \to \mathcal{B}(\mathcal{S} \times \mathcal{A})$ as

$$(T^*Q)(s, a) \triangleq r(s, a) + \gamma \int_\mathcal{S} \mathcal{P}(ds'|s, a) \max_{a'} Q(s', a').$$

The fixed point of the optimal Bellman operator is $(T^*Q^*) = Q^*$. As in Györfi et al. (2006), we define the truncation operator $\beta_B : \mathcal{B}(\mathcal{S} \times \mathcal{A}) \to \mathcal{B}(\mathcal{S} \times \mathcal{A}, B)$ such that, given a function $f \in \mathcal{B}(\mathcal{S} \times \mathcal{A})$, we have $\beta_B f(s, a) \in [-B, B], \forall (s, a) \in \mathcal{S} \times \mathcal{A}$.

## 3.2 APPROXIMATE VALUE ITERATION

The family of approximate value iteration (AVI) algorithms (Sutton & Barto, 2018; Farahmand, 2011; Munos, 2005) aims at approximating the solution of RL problems with large state space, where finding an optimal solution is impractical or impossible. Starting from an arbitrary estimate $Q^0$, AVI algorithms iteratively approximate the application of the optimal Bellman operator in a suitable function space, i.e., $Q^{k+1} \approx T^*Q^k$. Intuitively, the approximation error at each iteration plays a central role in terms of convergence to the fixed point $Q^*$. The approximation error results from two key aspects: (i) the unknown optimal Bellman operator, and (ii) the representation of the action-value function in a suitable function space $\mathcal{F}$ via the application of a nonlinear operator $S : \mathcal{B}(\mathcal{S} \times \mathcal{A}) \to \mathcal{F}$ defined as $Sy = \arg\inf_{f \in \mathcal{F}} \|f - y\|_\mu^2$ for any $y \in \mathcal{B}(\mathcal{S} \times \mathcal{A})$. Then, we define $Q^{k+1} = S(T^*Q^k)$, and the resulting approximation error by

$$\varepsilon_k \triangleq T^*Q^k - Q^{k+1}. \tag{1}$$

The role of the approximation error in AVI is evinced by the following theorem that bounds the performance loss, w.r.t. the optimal performance, at iteration $K$[1].

**Theorem 1.** *(Theorem 3.4 of Farahmand (2011)). Let $K$ be a positive integer, $\rho$ a distribution over states $s$, and $Q_{max} \leq \frac{R_{max}}{1-\gamma}$. Then, for any sequence $(Q_k)_{k=0}^K \subset \mathcal{B}(\mathcal{S} \times \mathcal{A}, Q_{max})$, and the corresponding sequence $(\varepsilon_k)_{k=0}^{K-1}$, we have*

$$\|Q^* - Q^K\|_{1,\rho} \leq \mathcal{G}(\gamma) \left[ A_{\rho,\mu}(K) \mathcal{E}^{1/2}(\varepsilon_0, \ldots, \varepsilon_{K-1}) + B(\gamma, K) R_{max} \right]$$

*where $\mathcal{E}(\varepsilon_0, \ldots, \varepsilon_{K-1}) = \sum_{k=0}^{K-1} C(k) \|\varepsilon_k\|_\mu^2$.*

We point out that the upper bound in Theorem 1 is a high-level version of the one reported in Farahmand (2011), where we omitted some technicalities that go beyond the scope of this paper replacing them with functions $\mathcal{G}$, $A$, $B$, and $C$. This simplified version highlights that the magnitude of the difference between the optimal action-value function $Q^*$ and the estimate at the $K$-th iteration $Q^K$, is proportional to the magnitude of the approximation errors $\varepsilon_k$ as defined in Equation 1. We refer the reader to the appendix for an extended version of the equation, and to Farahmand (2011) for further details about it.

---

[1]In Farahmand (2011), the action-value function at iteration $k$ is denoted by $Q^{\pi_k}$ to highlight its dependency on the policy $\pi_k$. In this paper, we simply write $Q^k$ omitting $\pi$ for simplicity.

## 4 CURRICULUM REINFORCEMENT LEARNING WITH BELLMAN RESIDUALS

We consider the classic curriculum RL scenario where an agent aims at learning a complex target task by solving a predefined sequence of easier, but increasingly challenging, versions of it (Narvekar et al., 2020). To avoid confusion on the debatable notion of difficulty of a task, in this work we adopt a practical point of view where the difficulty of a task is related to well-known aspects of RL, e.g., the complexity of the policies for solving them and challenging exploration (Florensa et al., 2017; Ivanovic et al., 2019; Klink et al., 2020).

### 4.1 APPROXIMATE VALUE ITERATION IN CURRICULUM REINFORCEMENT LEARNING

Let $\mathcal{T}_{1:n} = \{\mathcal{T}_1, \ldots, \mathcal{T}_n\}$ be a sequence of increasingly complex MDPs sharing the same state and action spaces, i.e., $\mathcal{T}_i = \langle \mathcal{S}, \mathcal{A}, \mathcal{P}_i, \mathcal{R}_i, \gamma \rangle$, and $\mathcal{T}_n$ be the target task. The use of AVI algorithms in this curriculum RL setting consists in starting from an arbitrary estimate $Q_1^0$ (where the subscript indicates the position of the task in the curriculum) and approximating the application of the task-specific optimal Bellman operator for the current task $Q_t^{k+1} \approx T_t^* Q_t^k$. We assume that the task changes when the estimate of the action-value of the current task converges. For simplicity, and without loss of generality, we denote $K_t$ the number of iterations needed to converge for each task $\mathcal{T}_t$, and denote $Q_t^0 = Q_{t-1}^{K_{t-1}} = Q_{t-1}^*$.

**Lemma 1.** *Let $K$ be the current iteration. Then,*

$$\|Q_n^* - Q_t^K\|_{1,\rho} \leq \|Q_n^* - Q_n^0\|_{1,\rho} + \cdots + \|Q_t^* - Q_t^K\|_{1,\rho} = \sum_{i=t+1}^{n} \|Q_i^* - Q_i^0\|_{1,\rho} + \|Q_t^* - Q_t^K\|_{1,\rho}. \quad (2)$$

This result can be verified applying the triangle inequality under the assumption that the algorithm switches task when the stopping condition of the algorithm is verified, e.g., when reaching a chosen number of iterations $K_t$ that allows the action-value function estimate to converge, or using heuristic methods checking the magnitude of the update (Lagoudakis & Parr, 2003; Ernst et al., 2005). Since zero is not a positive integer, the components in (2) cannot be bounded by the result in Theorem 1, which motivates our theoretical analysis in section 5. We denote the analytic $k$-step Bellman residual for task $t$ by

$$\bar{\varrho}_t^k \triangleq (T_t^*)^k Q_t^0 - Q_t^0 = (T_t^*)^k Q_{t-1}^* - Q_{t-1}^*. \quad (3)$$

**Proposition 1.** *Given an initial $Q_1^0(s,a)=0$ for any $(s,a) \in \mathcal{S} \times \mathcal{A}$, and considering that $Q_t^* = Q_t^{K_t} \approx (T_t^*)^{K_t} Q_{t-1}^*$, we obtain*

$$\bar{\varrho}_n^k + \sum_{t=1}^{n-1} \bar{\varrho}_t^{K_t} = (T_n^*)^k Q_n^0 - Q_n^0 + \left( \sum_{t=1}^{n-1} (T_t^*)^{K_t} Q_t^0 - Q_t^0 \right) \approx (T_t^*)^k \ldots (T_1^*)^{K_1} Q_1^0 - Q_1^0 \approx Q_n^k. \quad (4)$$

This result shows that, under above assumptions, the action-value function of the target task is approximately equivalent to the sum of $k$-step Bellman residuals. Notably, especially if the curriculum is sufficiently fine-grained, residuals are expected to become progressively smoother and smaller. In the following, we describe how to compute an estimate of the $k$-step residuals shown in (4).

### 4.2 BOOSTED CURRICULUM REINFORCEMENT LEARNING

It is well-known from supervised learning that: (i) small function approximators are easier to train, and thus are desirable to solve simple tasks; (ii) large function approximators can overfit or suffer slow training speed in simple tasks, but they are more suitable to handle complex problems. Ideally, we want to increase the representativeness of the function approximator with the increase of task complexity. Recalling (4), we propose to use a separate function approximator for each residual $\bar{\varrho}_t^k$, i.e. a separate approximator for each task. Given a dataset $\mathcal{D}_t = \{(s_t^i, a_t^i, r_t^i, s_t'^i)\}_{i=1}^N$ for each task, we denote the empirical Bellman operator by $(\hat{T}_t^* Q)(s_t^i, a_t^i) \triangleq r_t^i + \gamma \max_{a'} Q(s_t'^i, a')$, and define the $k$-step empirical residual as $\tilde{\varrho}_t^k \triangleq \hat{T}_t^* Q_t^{k-1} - Q_t^0$. At iteration $k$, for a current task $t$, and defining the nonlinear operator $\hat{S}\tilde{\varrho}_t^k = \arg\inf_{f \in \mathcal{F}} \|f - \tilde{\varrho}_t^k\|_{\mathcal{D}_t}^2$ as the empirical version of the operator $S$, we compute an approximation

$$Q_t^k = Q_t^0 + \beta_{B_k} \hat{S} \tilde{\varrho}_t^k = \sum_{i=1}^{t-1} \hat{\varrho}_i^{K_i} + \hat{\varrho}_t^k, \quad (5)$$

---

**Algorithm 1:** Boosted Curriculum Reinforcement Learning (BCRL)

---

**Input:** Tasks $\mathcal{T}_{1:n}$, $Q_1^0 = 0$;
**for** $t = 1, \ldots, n$ **do**

> Collect dataset $\mathcal{D}_t$ using policy induced by $Q_t^0$ (e.g., $\varepsilon$-greedy);
> **for** $k = 0, \ldots, K_t$ **do**
>
> > $\tilde{\varrho}_t^k \leftarrow \hat{T}_t^* Q_t^k - Q_t^0$;
> > $Q_t^{k+1} \leftarrow Q_t^0 + \beta_{B_k} \arg\inf_{f \in \mathcal{F}} \|f - \tilde{\varrho}_t^k\|_{\mathcal{D}_t}$;
>
> **end**
> $Q_{t+1}^0 \leftarrow Q_t^{K_t}$;

**end**
**Output:** $Q_n^{K_n} = Q_{n+1}^0 = \sum_{t=1}^n \hat{\varrho}_t^{K_t}$.

---

where we defined $\hat{\varrho}_t^k = \beta_{B_k} \hat{S} \tilde{\varrho}_t^k$ for compactness. For ease of notation, we use $Q_t^k$ also to denote the $Q$-function obtained by applying the empirical Bellman operator. We call this approach *boosted curriculum reinforcement learning* (BCRL), for its resemblance with the boosting procedure in supervised learning (Schapire, 1990; Freund, 1995), and we show its pseudocode in Algorithm 1.

**Remarks.** Our approach is based on the key insight that the strength of the function approximator should increase according to the complexity of the task at hand. In BCRL, we realize this increase in capacity by estimating an action-value function through a sum of action-value residuals of the previous tasks in the curriculum. Thus, contrary to methods using the same function space for all tasks, our approach has the advantage of acting in an augmented function space resulting from the sum of the function space of each approximator. In supervised learning literature, the function approximators in the sum are commonly known as weak regressors, meaning that boosting has the advantage of estimating complex functions as a sum of simple regressors (Geurts et al., 2006). Similarly, our BCRL procedure can use simple function approximators for each task in the curriculum, with the advantages that (i) easy tasks can be learned with easy function approximators curbing the risk of overfitting, (ii) the complexity of the function approximator can be chosen according to the individual tasks in the curriculum, contrarily to regular curriculum RL where the function approximator needs to be suitable for the target task. Conversely, at inference time, our method requires a forward pass for each approximator in the ensemble. Finally, we clarify that the soundness of this procedure is guaranteed by the fact that if the functional space of each weak regressor is Glivenko-Cantelli, i.e., "the error due to the empirical process goes to zero at least asymptotically", then, due to the preservation theorem, also the space resulting from the sum of weak regressors is Glivenko-Cantelli (Van Der Vaart & Wellner, 2000; Tosatto et al., 2017).

## 5 THEORETICAL ANALYSIS

We provide a theoretical analysis of BCRL in terms of convergence guarantees and finite-sample approximation error analysis. We refer to the well-known results in AVI (Farahmand, 2011; Farahmand et al., 2010), and the analysis provided in Tosatto et al. (2017), with which our analysis share some similarities.

### 5.1 CONVERGENCE GUARANTEES

**Theorem 2.** *Let $Q_{t-1}^*$ be the learned action-value for task $\mathcal{T}_{t-1}$, and $(Q_t^i)_{i=0}^k$ be a sequence of measurable action-value functions for a task $\mathcal{T}_t$ obtained following our BCRL procedure (5), where $Q_t^0 = Q_{t-1}^*$. Denote $L$ the Lipschitz coefficient of the optimal Bellman operator w.r.t. a norm $\|\cdot\|$, and assume the operator $S$ is such that $\exists \lambda > 0 : \|(I - S)y\| \leq \lambda \|y\|$ for all $y \in \mathcal{B}(\mathcal{S} \times \mathcal{A})$. Then,*

$$\|Q_t^k - Q_t^*\| \leq \left( \lambda \sum_{i=0}^{k-1} (L(1+\lambda))^i + (L(1+\lambda))^k \right) \|Q_t^0 - Q_t^*\|. \tag{6}$$

The assumption on the operator $S$ stated by Theorem 2 dictates that the approximation power of the used functional space should be good enough to learn the target function $y$; specifically, the upper bound of the approximation error is proportional to the magnitude of the function to approximate. This assumption holds for reasonable choices of function approximators and it is present in related works on the derivation of finite-time bounds for policy iteration (Munos, 2003), and value iteration (Munos, 2005; Tosatto et al., 2017). The following result shows that for each task, the interplay between $L$, $\lambda$ and $Q_t^0$ decides upon the approximation error w.r.t. $Q_t^*$.

**Corollary 1.** *Given the settings of Theorem 2, if $L < 1$ and $\lambda < \frac{\epsilon(1-L)}{1+L\epsilon}$ with $\epsilon \in [0,1]$ then*

$$\lim_{k\to\infty} \|Q_t^k - Q_t^*\| < \epsilon\|Q_t^0 - Q_t^*\|. \tag{7}$$

If now $Q_t^0 \approx Q_{t-1}^*$ is close to $Q_t^*$ in $L_p(\mu)$-norm, we can hope to achieve a small error in task $\mathcal{T}_t$.

### 5.2 FINITE-SAMPLE APPROXIMATION ERROR ANALYSIS

Given the inapplicability of Theorem 1 to the decomposition in Lemma 1, we now provide an alternative analysis for our BCRL method, in which the approximation errors for task $t$ translate to

$$\varepsilon_t^k = T_t^* Q_t^{k-1} - Q_t^k = \varrho_t^k - S\varrho_t^k, \qquad \varrho_t^k = T_t^* Q_t^{k-1} - Q_t^0. \tag{8}$$

**Theorem 3.** *(Theorem 5.3 of Farahmand (2011)) Let $(Q_t^i)_{i=0}^k$ be a sequence of action-value functions for each task $t$, $(\varepsilon_t^i)_{i=0}^k$ be the respective sequence of approximation errors as defined in (8), $\mathcal{F} \subseteq \mathcal{B}(\mathcal{S} \times \mathcal{A})$ be a subset of measurable action-value functions. Then we have*

$$\inf_{f\in\mathcal{F}} \|f - \varrho_t^k\|_\mu \le \inf_{f\in\mathcal{F}} \|f - \bar{\varrho}_t^k\|_\mu + \sum_{i=1}^{k-1} L^i \|\varepsilon_t^{k-i}\|_\mu. \tag{9}$$

This bound shows the relation between the approximation error of the residual for task $t$ at step $k$, the AVI Bellman Residual $\bar{\varrho}_t^k$, and the approximation error for previous residuals in task $t$. While the error is independent of previous tasks, the next Lemma shows a connection between tasks.

**Lemma 2.** *Let $(Q_t^i)_{i=0}^k$ be a sequence of action-value functions for each task $t$, $(\varepsilon_t^i)_{i=0}^k$ be the respective sequence of approximation errors as defined in (8). Then,*

$$\|\tilde{\varrho}_t^k\|_\infty \le \gamma\|\varepsilon_t^{k-1}\|_\infty + \gamma \sum_{i=1}^{k-2} \frac{2\gamma - (1+\gamma)\gamma^{k-1-i}}{1-\gamma} \|\varepsilon_t^i\|_\infty + R_t^{max} + \gamma \frac{1-\gamma^{k-1}}{1-\gamma} \|\varrho_t^1\|_\infty. \tag{10}$$

The above bound is based on the approximation errors $\varepsilon_t^i$ in task $\mathcal{T}_t$ as well as on the one-step bellman residual $\varrho_t^1 = T_t^* Q_t^0 - Q_t^0$. Intuitively, this residual will be small, if $Q_t^0 = Q_{t-1}^*$ is a good approximation to $Q_t^*$. We use (9) and (10) to finally bound the approximation error of the residual.

**Theorem 4.** *Let $B_t^k = \max(\|\tilde{\varrho}_t^k\|_\infty, 1)$, and $V_{\mathcal{F}^+}$ be the VC-dimension of $\mathcal{F}^+$, i.e., the class of all subgraphs of functions $f \in \mathcal{F}$ (Györfi et al., 2006). For a given sequence of action-value functions $(Q_t^i)_{i=0}^k$ generated by BCRL, we have*

$$\mathbb{E}\left[\|\varepsilon_t^k\|_\mu^2\right] \le \overbrace{4 \inf_{f\in\mathcal{F}} \|f - \bar{\varrho}_t^k\|_\mu^2}^{\text{(i) approximation error}} + \overbrace{4\Big(\sum_{i=1}^{k-1} L^i \|\varepsilon_t^{k-i}\|_\mu\Big)^2}^{\text{(ii) propagation error}}$$

$$+ \underbrace{\frac{5136 B_t^{k^4}}{N} \Big(\log 42e + 2\log(480e B_t^{k^2}) + 2\log(N)V_{\mathcal{F}^+}\Big)}_{\text{(iii) estimation error}}. \tag{11}$$

This result bounds the expected error of BCRL by a sum of three components: (i) the approximation error of the analytic residual $\bar{\varrho}_t^k$, (ii) the propagation error due to approximation errors at previous iterations, (iii) the estimation error due to the regression problem with a finite amount of samples. BCRL can be advantageous if the range of values of the residuals $\bar{\varrho}_t^k$ and $\tilde{\varrho}_t^k$ become smaller as the curriculum proceeds, which can be expected for a sufficiently fine-grained curriculum. In this case, fewer samples than for regular curriculum RL can be used to achieve comparable estimation errors, due to the smaller range of the target $\|\tilde{\varrho}_t^k\|_\mu$ compared to $Q_t^{\max}$.

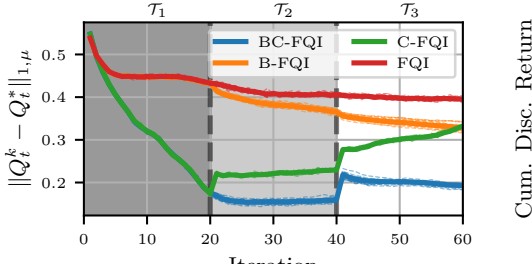 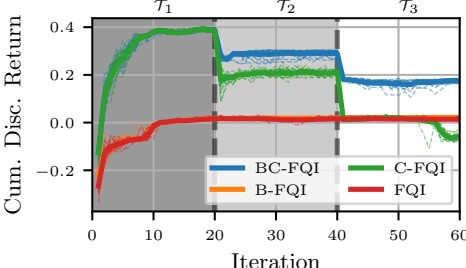

Figure 1: Mean absolute error (solid, bold lines) and standard error of the approximated $Q_t$-function over iterations as well as the corresponding performance obtained from a greedy policy w.r.t. this $Q_t$-function in task $\mathcal{T}_t$ in the car-on-hill environment (computed from 20 seeds). The differently shaded areas highlight the iterations on which BC-FQI and C-FQI train on tasks $\mathcal{T}_1$, $\mathcal{T}_2$ and $\mathcal{T}_3$. The fine, dashed lines indicate the results of the individual 20 runs from which the mean was computed.

## 6 EXPERIMENTS

This section serves to provide empirical evidence to the insights obtained in the previous section. More precisely, we want to provide evidence that boosted curricula can indeed improve the residual in an AVI setting and with that improve the achieved agent performance. Further, we show that the additional flexibility of boosting w.r.t. the individual function approximators can be leveraged to choose the best-suited learning approach for each learning task.

We conduct experiments in multiple environments and with two different RL algorithms using the MushroomRL library (D'Eramo et al., 2021).[2] To evaluate the benefit of curricula in boosted AVI settings, we turn towards the fitted $Q$-iteration (FQI) algorithm in the car-on-hill environment (Ernst et al., 2005), as the low-dimensionality allows for uniform sampling of the state-action space assumed in the previous section and the ground truth optimal $Q$-function can be computed via brute-force methods (Ernst et al., 2005). Next, we evaluate the FQI algorithm in a maze environment in which exploration is guided by the currently estimated optimal policy. Finally, we investigate boosting in a control task with DQN (Mnih et al., 2015), in which a point-mass needs to be steered over a narrow pathway subject to external perturbations. To disentangle the interplay between boosting and the use of a curriculum, we will compare different methods in the experiments: the algorithm under investigation (FQI, DQN), a version using both a curriculum with boosting (BC-FQI, BC-DQN) as well as two ablations of the second method - one without boosting (C-FQI, C-DQN) and one without a curriculum (B-FQI). The boosted methods that do not use a curriculum will introduce the additional approximators at the same iteration at which the curricula switches between tasks.

### 6.1 CAR-ON-HILL

The three car-on-hill tasks $\mathcal{T}_1$, $\mathcal{T}_2$, and $\mathcal{T}_3$ considered for curriculum learning in this experiment differ in the mass of the car, taking values of 0.8, 1.0, and 1.2 respectively. The default mass of the environment is 1.0 which means we are considering a version of this environment with more challenging exploration, as the car needs more energy to escape the valley and receive a reward signal. The dataset for learning the $Q$-function in each task is generated by executing 1000 trajectories with a random policy. To not make the results in this section dependent on the parametric form of the function approximator, we resort to the tree-based regression investigated in Ernst et al. (2005). The results are visualized in Figure 1. We can see that the combination of boosting and a curriculum (BC-FQI) outperforms all other methods both in terms of final approximation error achieved on $\mathcal{T}_3$ but also in terms of the performance of the greedy policy derived from the estimated $Q$-function. Comparing the residual of BC-FQI and C-FQI, we can see that C-FQI seems to be challenged with adapting the $Q$-function from $\mathcal{T}_1$ to $\mathcal{T}_2$, as the jump in residual shows. BC-FQI does not exhibit such a jump in residual when moving from $\mathcal{T}_1$ to $\mathcal{T}_2$. Furthermore, BC-FQI reduces the increased residual after switching from $\mathcal{T}_2$ to $\mathcal{T}_3$ while the residual of C-FQI steadily increases once arriving at the target task. We can particularly see that the pre-training in $\mathcal{T}_1$ and $\mathcal{T}_2$ leads to a significantly smaller $Q$-function error of BC-FQI compared to B-FQI at iteration 40, which is in line with the discussion in the previous sections.

---

[2]Code for reproducing results available under: `https://github.com/psclklnk/boosted_crl`. Additional experimental details can be found in the appendix.

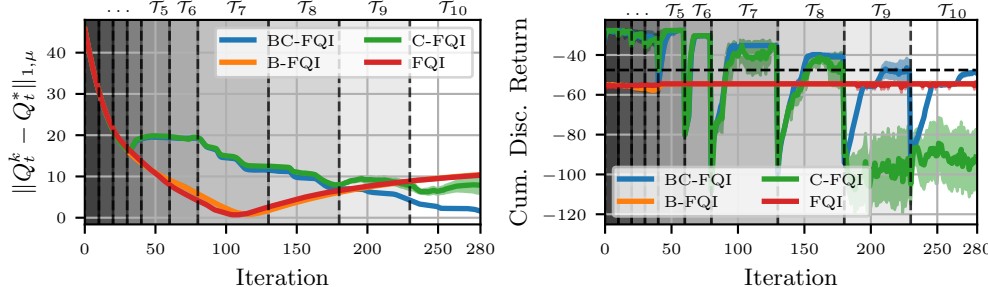

Figure 2: Average error in the approximated $Q_t$-functions (left) plus corresponding performance obtained from a greedy policy w.r.t. this $Q_t$-function (right) in the maze environment. Shaded colored areas correspond to two times standard error estimated from 40 seeds. The different shades of grey highlight the iterations in which BC-FQI and C-FQI train on different tasks $\mathcal{T}_t$. The horizontal black dashed line in the right plot highlights the maximum achievable reward in $\mathcal{T}_{10}$.

## 6.2 MAZE

We design a maze environment, in which an agent needs to reach a target position by navigating around two large obstacles. To reach the goal, the agent can choose between 4 discrete actions that move the agent up, down, left, or right. It obtains a dense reward based on the distance to the target and a reward of $+5$ upon reaching the goal. We design a curriculum composed of 10 tasks, in which the size of the walls gradually increases (see Figure 3a for a visualization of tasks contained in the curriculum). The reward function and the target task are designed to induce highly challenging exploration, requiring the use of curricula to solve the problem. As done for the car-on-hill experiment, we use FQI with tree-based regression (Ernst et al., 2005). For all methods, we collect a dataset for each task running 500 episodes using $\varepsilon$-greedy exploration, decreasing the strength of exploration as the curriculum proceeds. As can be seen in Figures 2 and 3, regular learning on the target task does not allow finding a good solution, regardless of whether boosting is used or not (B-FQI and FQI). The poor performance highlights the challenging exploration problem that needs to be solved in this environment. The initial random exploration is insufficient to maneuver around the second wall in the target task. Consequently, the agent is not aware of the benefit of this detour, as obvious in the learned $Q$-function (Figure 3b). Subsequently reducing the exploration does not lead to any new evidence and hence the agent quickly converges to the sub-optimal solution (Figure 2). Looking at the results of C-FQI, we see that the curriculum alleviates the aforementioned exploration problem. However, the estimated $Q$-function deteriorates before approaching the target task, leading to unreliable policies. By introducing additional capacity as learning proceeds, BC-FQI allows learning an optimal policy. Indeed, we show in the appendix that the residuals of BC-FQI decrease in later iterations, easing the function approximation problem when progressing to the target task.

## 6.3 LINEAR SYSTEM CONTROL

The last task aims at highlighting a more subtle benefit of BCRL: Choosing the architecture of the function approximators appropriately for each task in the curriculum. We show that this flexibility can improve sample efficiency in a control task, in which a point mass needs to be steered towards

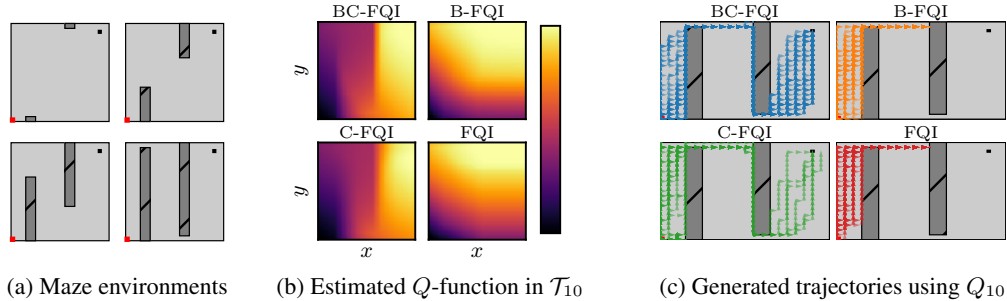

(a) Maze environments    (b) Estimated $Q$-function in $\mathcal{T}_{10}$    (c) Generated trajectories using $Q_{10}$

Figure 3: (a) visualizes different maze environments $\mathcal{T}_1, \mathcal{T}_4, \mathcal{T}_7$ and $\mathcal{T}_{10}$ (left to right, top to bottom). (b) shows the final $Q$-functions on $\mathcal{T}_{10}$ estimated by the evaluated approaches. (c) visualizes the trajectories generated by the corresponding agents.

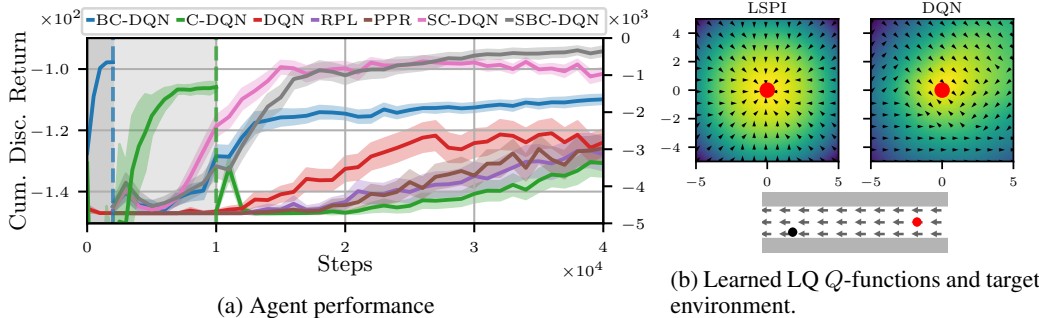

(a) Agent performance

(b) Learned LQ $Q$-functions and target environment.

Figure 4: (a) mean performance and standard error in the linear system control task achieved with different methods. The blue vertical dashed line indicates the iterations at which BC-DQN, RPL, PPR, SC-DQN and SBC-DQN switch from the LQ- to the target task. For C-DQN, this switch is indicated by the green dashed line. Statistics have been computed from 100 seeds. (b) visualizes the $Q$-functions learned during the pre-training in the LQ task with LSPI (for BC-DQN, RPL, PPR, SC-DQN and SBC-DQN) and DQN (for C-DQN). The arrows indicate the policy encoded by the $Q$-functions.

a goal position over a narrow pathway along which it is subject to a drift moving it away from the target position (Figure 4b). The agent receives a reward of $-1$ until reaching the goal, which issues a reward of $0$ and ends the episode. If the agent falls off the pathway, it is reset to the initial position. We learn a policy for solving this target task using DQN and introduce a curriculum that first trains in an easier system to induce a bias of moving towards the target. In this easier system, the point mass does not face external perturbations, obtains a reward equal to the negative squared distance to the target position, and does not need to stay on a particular pathway. Since the easier environment is a linear-quadratic (LQ) problem (see appendix C.3), the $Q$-function is known to be a quadratic function of state and action (Bradtke, 1993). The boosting framework allows to exploit this knowledge by using quadratic features in the first learning task and only relying on a neural network to realize the $Q$-function in the target task. With quadratic features, the $Q$-function can be efficiently learned from experience with least-squares policy iteration (LSPI) (Lagoudakis & Parr, 2003). In the boosted curriculum evaluated in this task (BC-DQN), we hence first learn a $Q$-function using LSPI in the LQ task and then adjust this $Q$-function in the target task with a residual using DQN.

Figure 4a shows that this incorporation of additional structure into the curriculum (BC-DQN) improves performance over both directly learning on the target task (DQN) and a regular curriculum that does not adjust the structure of the agent to the learning task in the curriculum (C-DQN). We further see that two other transfer knowledge baselines - Residual Policy Learning (RPL, Silver et al. (2018)) and Probabilistic Policy Reuse (PPR, Fernández et al. (2010)) - do not improve learning performance over regular learning on the target task (please see Appendix A for a discussion of the methods and why we chose them as baselines). Reusing the quadratic $Q$-function from the LQ problem as a shaping reward (SC-DQN, Ng et al. (1999)) drastically improves performance. Combined with boosting (SBC-DQN), this improvement is even stronger. It is particularly surprising that C-DQN, RPL, and PPR do not improve over DQN. For C-DQN, this poor performance may result from the DQN algorithm learning slower in the LQ problem compared to LSPI (10.000 vs. 2.000 steps), while further learning a less accurate $Q$-function (Figure 4b). For RPL and PPR, it is hard to provide explanations, as both methods rely on the LSPI $Q$-function. Nonetheless, the success of a shaping reward based on the same LSPI $Q$-function shows the impact of *how* existing knowledge is reused.

## 7 CONCLUSION

We introduced boosted curriculum reinforcement learning (BCRL), a new method for action-value function estimation in curriculum RL. For each task, BCRL approximates the respective action-value function as a sum of residuals of the previous tasks in the curriculum, increasing the representativeness of the functional space alongside the complexity of the tasks. We proved that BCRL enjoys advantageous theoretical properties in terms of convergence to the optimal action-value function of the target task. Moreover, we empirically validated BCRL across different problems and algorithms, showing its advantages over regular curriculum RL and knowledge transfer baselines. Future research on the presented findings should investigate ways to counteract the (linearly) growing complexity of the (boosted) agent for long task sequences and to adapt the BCRL framework to recent advances in automated curriculum generation (Narvekar et al., 2020).

## 8 REPRODUCIBILITY STATEMENT

We aim to guarantee the reproducibility of our work by providing the proofs of the main paper theorems and additional details of the conducted experiments in the appendix. Further, we provide the code for running the experiment at `https://github.com/psclklnk/boosted_crl`.

## 9 ACKNOWLEDGMENTS

This project has received funding from the DFG project PA3179/1-1 (ROBOLEAP).

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

## A  DISCUSSION OF RELATED TRANSFER LEARNING METHODS

Given our focus on knowledge transfer via $Q$-functions in curriculum RL, our investigations are closely related to the field of transfer learning and we hence want to situate our method in this field by this short review. Transfer learning is a well-established concept in RL and multiple surveys have been conducted in the single- and multi-agent RL setting (Taylor & Stone, 2009; Lazaric, 2012; Da Silva et al., 2018; Da Silva & Costa, 2019). Apart from detecting a change in the current task (Hernandez-Leal et al., 2017), determining from which task to transfer knowledge (Fernández & Veloso, 2006; Barrett & Stone, 2015; Hernandez-Leal & Kaisers, 2017) to the current task is an important problem in transfer learning. For an adequately chosen curriculum, the aforementioned problems are less severe as a) the agent is assumed to be aware of the task change and b) the knowledge from the most recent task in the curriculum can be assumed to be the most relevant for solving the current one. The transfer of knowledge between tasks can then happen in various forms, e.g., via samples (Lazaric et al., 2008; Lazaric & Restelli, 2011), policies (Fernández & Veloso, 2006; Fernández et al., 2010; Silver et al., 2018), reward functions (Brys et al., 2015) or demonstrations/teachers (Chernova & Veloso, 2009; Billard et al., 2016; Koert et al., 2020).

When sharing value- or $Q$-functions, which resembles the boosting scenario investigated in this paper, incorporating abstract concepts like objects, classes, or relations (Koga et al., 2013; 2014; Da Silva & Costa, 2017; Silva & Costa, 2018) can drastically improve the generalization of learned $Q$- or value-functions between tasks at the expense of additional user-specified information. When not assuming any additional information of the aforementioned forms in a value- or $Q$-function transfer setting, methods that extend upon the naive idea of initializing the $Q$-function of the next task with the most recently learned one, as e.g. done in Narvekar et al. (2016), are rather sparse. Boutsioukis et al. (2011) investigate the idea of sparsely initializing the $Q$-function in the new task with a bias value for the optimal actions of the previous task in discrete state-action spaces. Rusu et al. (2016) present a particular neural network architecture – progressive neural networks – that aims to improve transfer between tasks by adding so-called columns for each encountered task. Those additional columns are connected to the columns of the previous task by learnable weight matrices. Reward shaping (Ng et al., 1999) allows reusing a value function from a source task as a shaping function in a target task. Both progressive neural networks and reward shaping are orthogonal to our investigation of boosting and its influence on the residual of $Q$-function approximators in curriculum RL. As we show in Figure 4a, boosting hence can still provide benefits when combined with such approaches.

For the comparison in Section 6.3, we choose residual policy learning (RPL, Silver et al. (2018)), as it follows a similar idea of learning a residual, however in the form of a policy, and avoids the complication of the overall curriculum by e.g. introducing a secondary RL objective (Brys et al., 2015) to realize the knowledge transfer between tasks or the need to choose appropriate bias values for $Q$-function initialization (Boutsioukis et al., 2011). For the environment in Section 6.3, the idea of a residual policy is well-suited since the 8 discrete actions in the target task (see appendix C.3) allow to define reasonable ways to combine the actions of the previous and the current policy. For the tasks in Section 6.1 and 6.2, combining actions is problematic due to the action spaces of the tasks. For the car-on-hill task, combining the two opposing actions of moving left or right will either always ignore the actions of the previous policy or ignore the actions of the current policy. The same problem occurs for the maze task with its four actions *left*, *right*, *up*, and *down*. Apart from RPL, we also evaluate probabilistic policy reuse (PPR, Fernández et al. (2010)) as it – like RPL– introduces no crucial additional design choices in the curriculum.

## B  ADDITIONAL THEOREMS AND PROOFS

We report an extended version of our Theorem 1 based on Theorem 3.4 in Farahmand (2011).

**Theorem 1.** *(Theorem 3.4 of Farahmand (2011)). Let $K$ be a positive integer, $\rho$ an initial state distribution, and $Q_{max} \leq \frac{R_{max}}{1-\gamma}$. Then, for any sequence $(Q_k)_{k=0}^{K} \subset \mathcal{B}(\mathcal{S} \times \mathcal{A}, Q_{max})$, and the corresponding sequence $(\varepsilon_k)_{k=0}^{K-1}$, we have*

$$\|Q^* - Q^K\|_{1,\rho} \leq \frac{2\gamma}{(1-\gamma)^2} \left[ \inf_{r \in [0,1]} C_{VI,\rho,\mu}^{\frac{1}{2}}(K;r) \mathcal{E}^{\frac{1}{2}}(\varepsilon_0, \ldots, \varepsilon_{K-1}; r) + \frac{2}{1-\gamma} \gamma^K R_{max} \right]$$

*where $\mathcal{E}(\varepsilon_0, \ldots, \varepsilon_{K-1}; r) = \sum_{k=0}^{K-1} \alpha_k^{2r} \|\varepsilon_k\|_\mu^2$, and $C_{VI,\rho,\mu}$ and $\alpha_k$ as defined, respectively, in Definition 3.1 and Equation 3.3 in Farahmand (2011).*

We proceed reporting the proofs of the theorems and lemmas described in the main paper. As a start, we first restate the most important identities:

$$\bar{\varrho}_t^k = (T_t^*)^k Q_t^0 - Q_t^0 \qquad \varrho_t^k = T_t^* Q_t^{k-1} - Q_t^0 \qquad \varrho_t^1 = T_t^* Q_t^0 - Q_t^0$$

$$Q_t^k = Q_t^0 + S\varrho_t^k \qquad \varepsilon_t^k = \varrho_t^k - S\varrho_t^k \qquad Q_t^k + \varepsilon_t^k = T_t^* Q_t^{k-1}$$

**Theorem 2.** *Let $Q_{t-1}^*$ be the learned action-value for task $\mathcal{T}_{t-1}$, and $(Q_t^i)_{i=0}^k$ be a sequence of measurable action-value functions for a task $\mathcal{T}_t$ obtained following our BCRL procedure (5), where $Q_t^0 = Q_{t-1}^*$. Denote $L$ the Lipschitz coefficient of the optimal Bellman operator w.r.t. a norm $\|\cdot\|$, and assume the operator $S$ is such that $\exists \lambda > 0 : \|(I - S)y\| \le \lambda\|y\|$ for all $y \in \mathcal{B}(\mathcal{S} \times \mathcal{A})$. Then,*

$$\|Q_t^k - Q_t^*\| \le \left( \lambda \sum_{i=0}^{k-1} (L(1+\lambda))^i + (L(1+\lambda))^k \right) \left\| Q_t^0 - Q_t^* \right\|. \tag{12}$$

*Proof.* Before carrying out the proof, we want to clarify the role of the chosen norm $\|\cdot\|$. While it is well known that the Bellman operator is a contraction for the supremum norm $\|\cdot\|_\infty$ (since $\|T^* Q_1 - T^* Q_2\|_\infty \le \gamma\|Q_1 - Q_2\|_\infty$ for any $Q_1, Q_2 \in \mathcal{B}(\mathcal{S} \times \mathcal{A})$), the use of the supremum norm puts a tougher requirement on the chosen function approximator $S$, as for the supremum norm the largest approximation error determines $\lambda$. The use of the $L_p(\mu)$-norms eases the aforementioned requirement on $S$, however, for the $L_p(\mu)$-norm it is not guaranteed that the Bellman operator is a contraction for any $Q_1, Q_2 \in \mathcal{B}(\mathcal{S} \times \mathcal{A})$. While Farahmand (2011) upper bounded the Lipschitz coefficient of the optimal Bellman operator for $L_p(\mu)$-norms by

$$\gamma C_{\mu \to \infty} = \gamma \left( \int_{\mathcal{S} \times \mathcal{A}} d\mu(s, a) \sup_{(s', a') \in \mathcal{S} \times \mathcal{A}} \frac{1}{\pi_b(a'|s')} \frac{dP^{s,a}}{d\mu_{\mathcal{S}}}(s') \right)^{\frac{1}{2}}, \tag{13}$$

where $\mu(s, a) = \mu_{\mathcal{S}}(s)\pi_b(a|s)$, $\gamma C_{\mu \to \infty}$ is typically larger than one. Note, however, that this is only an upper-bound and grid-world simulations indicated that $L < 1$ for many $Q_1, Q_2 \in \mathcal{B}(\mathcal{S} \times \mathcal{A})$, however, not all. While AVI convergence results in $L_p(\mu)$-norm such as (Munos, 2005) obviously apply to BCRL, the bound in Theorem 2 particularly highlights the dependency on the $Q$-function learned in task $t - 1$. After this discussion, we now prove the actual theorem:

$$\left\| Q_t^k - Q_t^* \right\| \le \left\| Q_t^k - T_t^* Q_t^{k-1} \right\| + \left\| T_t^* Q_t^{k-1} - Q_t^* \right\|$$

$$= \left\| \varepsilon_t^k \right\| + \left\| T_t^* Q_t^{k-1} - Q_t^* \right\|$$

$$\le \left\| \varrho_t^k - S\varrho_t^k \right\| + L\left\| Q_t^{k-1} - Q_t^* \right\|$$

$$\le \lambda\left\| \varrho_t^k \right\| + L\left\| Q_t^{k-1} - Q_t^* \right\| \tag{14}$$

$$\le \lambda\left\| Q_t^0 - Q_t^* \right\| + L(1+\lambda)\left\| Q_t^{k-1} - Q_t^* \right\|, \tag{15}$$

where (14) considers the assumption on the operator $(I - S)$ and (15) follows noting that

$$\left\| \varrho_t^k \right\| \le \left\| T_t^* Q_t^{k-1} - T_t^* Q_t^* \right\| + \left\| T_t^* Q_t^* - Q_t^0 \right\| \le L\left\| Q_t^{k-1} - Q_t^* \right\| + \left\| Q_t^0 - Q_t^* \right\|.$$

Unrolling the recursion in (15) then proves the desired result:

$$\left\| Q_t^k - Q_t^* \right\| \le \lambda\left\| Q_t^0 - Q_t^* \right\| + L(1+\lambda)\left\| Q_t^{k-1} - Q_t^* \right\|$$

$$\le \lambda\left\| Q_t^0 - Q_t^* \right\| + L(1+\lambda)\left( \lambda\left\| Q_t^0 - Q_t^* \right\| + L(1+\lambda)\left\| Q_t^{k-2} - Q_t^* \right\| \right)$$

$$= \lambda\left( \left\| Q_t^0 - Q_t^* \right\| + L(1+\lambda)\left\| Q_t^0 - Q_t^* \right\| \right) + (L(1+\lambda))^2\left\| Q_t^{k-2} - Q_t^* \right\|$$

$$\le \dots \le \lambda \sum_{i=0}^{k-1} (L(1+\lambda))^i\left\| Q_t^0 - Q_t^* \right\| + (L(1+\lambda))^k\left\| Q_t^0 - Q_t^* \right\|.$$

$\square$

**Corollary 1.** *Given the settings of Theorem 2, if $L < 1$ and $\lambda < \frac{\epsilon(1-L)}{1+L\epsilon}$ with $\epsilon \in [0,1]$ then*

$$\lim_{k \to \infty} \|Q_t^k - Q_t^*\| < \epsilon \|Q_t^0 - Q_t^*\|.$$

*Proof.* If we assume $\lambda = \frac{1-L}{L} - \delta$ with $0 < \delta \leq \frac{1-L}{L}$, then it follows that $L(1 + \lambda) = 1 - L\delta < 1$ and hence

$$\lim_{k \to \infty} \left( \lambda \sum_{i=0}^{k-1} (L(1+\lambda))^i + (L(1+\lambda))^k \right)$$

$$= \lim_{k \to \infty} \left( \lambda \sum_{i=0}^{k-1} (1 - L\delta)^i + (1 - L\delta)^k \right)$$

$$= \lambda \lim_{k \to \infty} \sum_{i=0}^{k-1} (1 - L\delta)^i + \lim_{k \to \infty} (1 - L\delta)^k$$

$$= \lambda \frac{1}{1 - (1 - L\delta)} = \left( \frac{1-L}{L} - \delta \right) \frac{1}{L\delta} = \frac{1 - L - L\delta}{L^2\delta}.$$

As a next step, we require that the above term is upper bounded by a factor $\epsilon \in [0,1]$. This requirement yields

$$\lim_{k \to \infty} \left( \lambda \sum_{i=0}^{k-1} (L(1+\lambda))^i + (L(1+\lambda))^k \right) = \frac{1 - L - L\delta}{L^2\delta} < \epsilon$$

$$\Leftrightarrow 1 - L - L\delta < L^2\delta\epsilon$$

$$\Leftrightarrow 1 - L < \delta(L + L^2\epsilon)$$

$$\Leftrightarrow \frac{1-L}{L + L^2\epsilon} < \delta$$

Note that it holds that $0 < \delta = \frac{1-L}{L+L^2\epsilon} \leq \frac{1-L}{L}$ for any $\epsilon \in [0,1]$ such that our initial assumption on $\lambda$ are fulfilled. Inserting this inequality into our definition of $\lambda$ then finally yields

$$\lambda = \frac{1-L}{L} - \delta \Leftrightarrow \lambda < \frac{1-L}{L} - \frac{1-L}{L + L^2\epsilon}$$

$$= \frac{(1-L)(1+L\epsilon) - (1-L)}{L + L^2\epsilon}$$

$$= \frac{1 + L\epsilon - L - L^2\epsilon - 1 + L}{L + L^2\epsilon}$$

$$= \frac{L\epsilon - L^2\epsilon}{L + L^2\epsilon} = \frac{\epsilon - L\epsilon}{1 + L\epsilon} = \frac{\epsilon(1-L)}{1 + L\epsilon}$$

$\square$

**Theorem 3.** *(Theorem 5.3 of Farahmand (2011)) Let $(Q_t^i)_{i=0}^k$ be a sequence of action-value functions for each task $t$, $(\varepsilon_t^i)_{i=0}^k$ be the respective sequence of approximation errors as defined in (8), $\mathcal{F} \subseteq \mathcal{B}(\mathcal{S} \times \mathcal{A})$ be a subset of measurable action-value functions. Then we have*

$$\inf_{f \in \mathcal{F}} \|f - \varrho_t^k\|_\mu \leq \inf_{f \in \mathcal{F}} \|f - \bar{\varrho}_t^k\|_\mu + \sum_{i=1}^{k-1} L^i \|\varepsilon_t^{k-i}\|_\mu. \tag{16}$$

*Proof.* Although this theorem is the same as Theorem 5.3 in (Farahmand, 2011), we still provide the proof here for ease of understanding. For an arbitrary function $f \in \mathcal{F}$ and task $t$, we have $\|f - \varrho_t^k\|_\mu \leq \|f - \bar{\varrho}_t^k\|_\mu + \|\bar{\varrho}_t^k - \varrho_t^k\|_\mu$. Then,

$$\left\| \bar{\varrho}_t^k - \varrho_t^k \right\|_\mu = \left\| (T_t^*)^k Q_t^0 - T_t^* Q_t^{k-1} \right\|_\mu$$

$$\leq L\left\|(T_t^*)^{k-1}Q_t^0 - Q_t^{k-1}\right\|_\mu$$

$$= L\left\|(T_t^*)^{k-1}Q_t^0 - T_t^*Q_t^{k-2} + T_t^*Q_t^{k-2} - Q_t^{k-1}\right\|_\mu$$

$$= L\left\|(T_t^*)^{k-1}Q_t^0 - T_t^*Q_t^{k-2} + \varepsilon_t^{k-1}\right\|_\mu$$

$$\leq L\left(\left\|\varepsilon_t^{k-1}\right\|_\mu + \left\|(T_t^*)^{k-1}Q_t^0 - T_t^*Q_t^{k-2}\right\|_\mu\right)$$

$$\leq L\left(\left\|\varepsilon_t^{k-1}\right\|_\mu + \left\|\bar{\varrho}_t^{k-1} - \varrho_t^{k-1}\right\|_\mu\right)$$

$$\leq L\left(\left\|\varepsilon_t^{k-1}\right\|_\mu + L\left(\left\|\varepsilon_t^{k-2}\right\|_\mu + \left\|\bar{\varrho}_t^{k-2} - \varrho_t^{k-2}\right\|_\mu\right)\right)$$

$$\leq \ldots \leq \sum_{i=1}^{k-1} L^i\|\varepsilon_t^{k-i}\|_\mu + L^{k-1}\|\bar{\varrho}_t^1 - \varrho_t^1\|_\mu = \sum_{i=1}^{k-1} L^i\|\varepsilon_t^{k-i}\|_\mu.$$

For the last equality, we used that $\|\bar{\varrho}_t^1 - \varrho_t^1\|_\mu = \|T_t^*Q_t^0 - T_t^*Q_t^0\|_\mu = 0.$ $\qquad\square$

**Theorem 4.** *Let $B_t^k = \max(\|\tilde{\varrho}_t^k\|_\infty, 1)$, and $V_{\mathcal{F}^+}$ be the VC-dimension of $\mathcal{F}^+$, which is the class of all subgraphs of functions $f \in \mathcal{F}$ (Györfi et al., 2006). Given a sequence of action-value functions $(Q_t^i)_{i=0}^k$ generated by our BCRL, we have*

$$\mathbb{E}\left[\|\varepsilon_t^k\|_\mu^2\right] \leq 4\inf_{f \in \mathcal{F}}\|f - \bar{\varrho}_t^k\|_\mu^2 + 4\left(\sum_{i=1}^{k-1} L^i\|\varepsilon_t^{k-i}\|_\mu\right)^2$$

$$+ \frac{5136{B_t^k}^4}{N}\left(\log 42e + 2\log(480e{B_t^k}^2) + 2\log(N)V_{\mathcal{F}^+}\right). \tag{17}$$

*Proof.* Using Theorem 11.5 from Györfi et al. (2006), we obtain

$$\mathbb{E}\left[\|\varepsilon_t^k\|_\mu^2\right] \leq 2\inf_{f \in \mathcal{F}}\|f - \varrho_t^k\|_\mu^2 + \frac{5136{B_t^k}^4}{N}\left(\log 42e + 2\log(480e{B_t^k}^2) + 2\log(N)V_{\mathcal{F}^+}\right).$$

Completing the proof requires to relate $\|f - \varrho_t^k\|_\mu^2$ to (16). This is achieved using the Cauchy-Schwarz inequality:

$$\|f - \varrho_t^k\|_\mu^2 \leq \left(\|f - \bar{\varrho}_t^k\|_\mu + \|\bar{\varrho}_t^k - \varrho_t^k\|_\mu\right)^2$$

$$\leq 2\|f - \bar{\varrho}_t^k\|_\mu^2 + 2\|\bar{\varrho}_t^k - \varrho_t^k\|_\mu^2 \leq 2\|f - \bar{\varrho}_t^k\|_\mu^2 + 2\left(\sum_{i=1}^{k-1} L^i\|\varepsilon_t^{k-i}\|_\mu\right)^2.$$

Plugging above inequality into the result obtained from Györfi et al. (2006) concludes the proof. $\quad\square$

**Lemma 2.** *Let $(Q_t^i)_{i=0}^k$ be a sequence of action-value functions for each task $t$, $(\varepsilon_t^i)_{i=0}^k$ be the respective sequence of approximation errors as defined in (8). Then,*

$$\|\tilde{\varrho}_t^k\|_\infty \leq \gamma\|\varepsilon_t^{k-1}\|_\infty + \gamma\sum_{i=1}^{k-2}\frac{2\gamma - (1+\gamma)\gamma^{k-1-i}}{1-\gamma}\|\varepsilon_t^i\|_\infty + R_t^{max} + \gamma\frac{1-\gamma^{k-1}}{1-\gamma}\|\varrho_t^1\|_\infty. \tag{18}$$

*Proof.* As a first step, we bound the $k$-step Bellman residual using the following Lemma.

**Lemma 3.** *Let $(Q_t^i)_{i=0}^k$ be a sequence of action-value functions for each task $t$, $(\varepsilon_t^i)_{i=0}^k$ be the respective sequence of approximation errors as defined in (8). Then,*

$$\|\varrho_t^k\|_\infty \leq \sum_{i=1}^{k-1}\frac{2\gamma - (1+\gamma)\gamma^{k-i}}{1-\gamma}\|\varepsilon_t^i\|_\infty + \frac{1-\gamma^k}{1-\gamma}\|\varrho_t^1\|_\infty. \tag{19}$$

*Proof.*

$$\|\varrho_t^k\|_\infty = \|T_t^* Q_t^{k-1} - Q_t^{k-1} + Q_t^{k-1} - Q_t^0\|_\infty$$

$$= \|T_t^* Q_t^{k-1} - Q_t^{k-1} + T_t^* Q_t^{k-2} - \varepsilon_t^{k-1} - Q_t^0\|_\infty$$

$$= \|T_t^* Q_t^{k-1} - Q_t^{k-1} - \varepsilon_t^{k-1} + T_t^* Q_t^{k-2} - Q_t^0\|_\infty$$

$$= \Big\| \sum_{i=1}^{k-1} (T_t^* Q_t^i - Q_t^i - \varepsilon_t^i) + \varrho_t^1 \Big\|_\infty$$

$$= \sup_{s\in\mathcal{S}, a\in\mathcal{A}} \Big| r_t(s,a) + \gamma \int_\mathcal{S} \mathcal{P}_t(ds'|s,a) \max_{a'\in\mathcal{A}} Q_t^{k-1}(s',a') - Q_t^{k-1}(s,a) - \varepsilon_t^{k-1}(s,a)$$

$$+ \sum_{i=1}^{k-2} (T_t^* Q_t^i - Q_t^i + \varepsilon_t^i) + \varrho_t^1 \Big|$$

$$= \sup_{s\in\mathcal{S}, a\in\mathcal{A}} \Big| r_t(s,a) + \gamma \int_\mathcal{S} \mathcal{P}_t(ds'|s,a) \max_{a'\in\mathcal{A}} Q_t^{k-1}(s',a') - Q_t^{k-1}(s,a) - \varepsilon_t^{k-1}(s,a) + \dots \Big|$$

$$= \sup_{s\in\mathcal{S}, a\in\mathcal{A}} \Big| r_t(s,a) + \gamma \int_\mathcal{S} \mathcal{P}_t(ds'|s,a) \max_{a'\in\mathcal{A}} Q_t^{k-1}(s',a') - T_t^* Q_t^{k-2}(s,a) + \dots \Big|$$

$$= \sup_{s\in\mathcal{S}, a\in\mathcal{A}} \Big| \gamma \int_\mathcal{S} \mathcal{P}_t(ds'|s,a) \Big( \max_{a'\in\mathcal{A}} Q_t^{k-1}(s',a') - \max_{a'\in\mathcal{A}} Q_t^{k-2}(s',a') \Big) + \dots \Big|$$

$$= \sup_{s\in\mathcal{S}, a\in\mathcal{A}} \Big| \gamma \int_\mathcal{S} \mathcal{P}_t(ds'|s,a) \Big( \max_{a'\in\mathcal{A}}(T_t^* Q_t^{k-2}(s',a') - \varepsilon_t^{k-1}(s',a')) - \max_{a'\in\mathcal{A}} Q_t^{k-2}(s',a') \Big) + \dots \Big|$$

$$\leq \gamma \left( \|\varepsilon_t^{k-1}\|_\infty + \|T_t^* Q_t^{k-2} - Q_t^{k-2}\|_\infty \right) + \Big\| \sum_{i=1}^{k-2} (T_t^* Q_t^i - Q_t^i - \varepsilon_t^i) + \varrho_t^1 \Big\|_\infty$$

$$\leq \gamma \sum_{i=1}^{k-1} \left( \|\varepsilon_t^i\|_\infty + \|T_t^* Q_t^{i-1} - Q_t^{i-1}\|_\infty \right) + \|\varrho_t^1\|_\infty. \tag{20}$$

We need to bound the individual terms $\|T_t^* Q_t^{i-1} - Q_t^{i-1}\|_\infty$. We proceed as Tosatto et al. (2017):

$$\|T_t^* Q_t^k - Q_t^k\|_\infty$$

$$= \sup_{s\in\mathcal{S}, a\in\mathcal{A}} \Big| r(s,a) + \gamma \int_\mathcal{S} \mathcal{P}_t(ds'|s,a) \max_{a'\in\mathcal{A}} Q_t^k(s',a') - Q_t^k(s,a) \Big|$$

$$= \sup_{s\in\mathcal{S}, a\in\mathcal{A}} \Big| r(s,a) + \gamma \int_\mathcal{S} \mathcal{P}_t(ds'|s,a) \max_{a'\in\mathcal{A}}(T_t^* Q_t^{k-1}(s',a') - \varepsilon_t^k(s',a')) - (T_t^* Q_t^{k-1}(s,a) - \varepsilon_t^k(s,a)) \Big|$$

$$\leq (1+\gamma)\|\varepsilon_t^k\|_\infty + \gamma \sup_{s\in\mathcal{S}, a\in\mathcal{A}} \Big| \int_\mathcal{S} \mathcal{P}_t(ds'|s,a) \Big( \max_{a'\in\mathcal{A}}(T_t^* Q_t^{k-1}(s',a') - \max_{a'\in\mathcal{A}} Q_t^{k-1}(s',a') \Big) \Big|$$

$$\leq (1+\gamma)\|\varepsilon_t^k\|_\infty + \gamma \|T_t^* Q_t^{k-1} - Q_t^{k-1}\|_\infty$$

$$\leq (1+\gamma)\left( \|\varepsilon_t^k\|_\infty + \gamma \|\varepsilon_t^{k-1}\|_\infty \right) + \gamma^2 \|T_t^* Q_t^{k-2} - Q_t^{k-2}\|_\infty$$

$$\leq \dots \leq (1+\gamma) \sum_{i=0}^{k-1} \gamma^i \|\varepsilon_t^{k-i}\|_\infty + \gamma^k \|T_t^* Q_t^0 - Q_t^0\|_\infty$$

$$= (1+\gamma) \sum_{i=1}^{k} \gamma^{k-i} \|\varepsilon_t^i\|_\infty + \gamma^k \|\varrho_t^1\|_\infty \tag{21}$$

Inserting (21) into (20) then yields

$$\|\varrho_t^k\|_\infty \leq \gamma \sum_{i=1}^{k-1} \left( \|\varepsilon_t^i\|_\infty + \|T_t^* Q_t^{i-1} - Q_t^{i-1}\|_\infty \right) + \|\varrho_t^1\|_\infty$$

$$\leq \gamma \sum_{i=1}^{k-1} \left( \|\varepsilon_t^i\|_\infty + (1+\gamma) \sum_{j=1}^{i-1} \gamma^{i-1-j} \|\varepsilon_t^j\|_\infty + \gamma^{i-1} \|\varrho_t^1\|_\infty \right) + \|\varrho_t^1\|_\infty$$

$$
\begin{aligned}
&= \sum_{i=1}^{k-1} \left( \gamma \|\varepsilon_t^i\|_\infty + (1+\gamma) \sum_{j=1}^{i-1} \gamma^{i-j} \|\varepsilon_t^j\|_\infty \right) + \sum_{i=0}^{k-1} \gamma^i \|\varrho_t^1\|_\infty \\
&= \sum_{i=1}^{k-1} \gamma \|\varepsilon_t^i\|_\infty + (1+\gamma) \sum_{i=1}^{k-1} \sum_{j=1}^{i-1} \gamma^{i-j} \|\varepsilon_t^j\|_\infty + \frac{1-\gamma^k}{1-\gamma} \|\varrho_t^1\|_\infty \\
&= \sum_{i=1}^{k-1} \gamma \|\varepsilon_t^i\|_\infty + (1+\gamma) \sum_{i=1}^{k-1} \left( \sum_{j=1}^{i} \gamma^{i-j} \|\varepsilon_t^j\|_\infty - \|\varepsilon_t^i\|_\infty \right) + \frac{1-\gamma^k}{1-\gamma} \|\varrho_t^1\|_\infty \\
&= \sum_{i=1}^{k-1} (\gamma - (1+\gamma)) \|\varepsilon_t^i\|_\infty + (1+\gamma) \sum_{j=1}^{k-1} \sum_{i=j}^{k-1} \gamma^{i-j} \|\varepsilon_t^j\|_\infty + \frac{1-\gamma^k}{1-\gamma} \|\varrho_t^1\|_\infty \\
&= -\sum_{i=1}^{k-1} \|\varepsilon_t^i\|_\infty + (1+\gamma) \sum_{j=1}^{k-1} \|\varepsilon_t^j\|_\infty \sum_{i=j}^{k-1} \gamma^{i-j} + \frac{1-\gamma^k}{1-\gamma} \|\varrho_t^1\|_\infty \\
&= -\sum_{i=1}^{k-1} \|\varepsilon_t^i\|_\infty + (1+\gamma) \sum_{j=1}^{k-1} \|\varepsilon_t^j\|_\infty \sum_{i=0}^{k-j-1} \gamma^{i} + \frac{1-\gamma^k}{1-\gamma} \|\varrho_t^1\|_\infty \\
&= -\sum_{i=1}^{k-1} \|\varepsilon_t^i\|_\infty + (1+\gamma) \sum_{j=1}^{k-1} \frac{1-\gamma^{k-j}}{1-\gamma} \|\varepsilon_t^j\|_\infty + \frac{1-\gamma^k}{1-\gamma} \|\varrho_t^1\|_\infty \\
&= \sum_{i=1}^{k-1} \frac{(1+\gamma)(1-\gamma^{k-i}) - (1-\gamma)}{1-\gamma} \|\varepsilon_t^i\|_\infty + \frac{1-\gamma^k}{1-\gamma} \|\varrho_t^1\|_\infty
\end{aligned}
$$

The proof is completed by reformulating $(1+\gamma)(1-\gamma^{k-i}) - (1-\gamma) = 2\gamma - (1+\gamma)\gamma^{k-i}$. $\qquad\square$

Equipped with the result from Lemma 3, we are now ready to prove Lemma 2 by unrolling $\|\tilde{\varrho}_t^k\|_\infty$ and using (19):

$$
\begin{aligned}
\|\tilde{\varrho}_t^k\|_\infty &= \|\hat{T}_t^* Q_t^{k-1} - Q_t^0\|_\infty \\
&= \sup_{(s,a,r,s') \in \mathcal{D}_t} \left| r + \gamma \max_{a' \in \mathcal{A}} Q_t^{k-1}(s',a') - Q_t^0(s,a) \right| \\
&= \sup_{(s,a,r,s') \in \mathcal{D}_t} \left| r + \gamma \max_{a' \in \mathcal{A}} \left( T_t^* Q_t^{k-2}(s',a') - \varepsilon_t^{k-1}(s',a') \right) - Q_t^0(s,a) \right| \\
&\leq \gamma \|\varepsilon_t^{k-1}\|_\infty + R_t^{\max} + \gamma \sup_{s'' \in \mathcal{S}} \max_{a' \in \mathcal{A}} \left| T_t^* Q_t^{k-2}(s'',a') - Q_t^0(s'',a') \right| \\
&\leq \gamma \|\varepsilon_t^{k-1}\|_\infty + R_t^{\max} + \gamma \|\varrho_t^{k-1}\|_\infty \\
&\leq \gamma \|\varepsilon_t^{k-1}\|_\infty + \gamma \sum_{i=1}^{k-2} \frac{2\gamma - (1+\gamma)\gamma^{k-1-i}}{1-\gamma} \|\varepsilon_t^i\|_\infty + R_t^{\max} + \gamma \frac{1-\gamma^{k-1}}{1-\gamma} \|\varrho_t^1\|_\infty .
\end{aligned}
$$

$\qquad\square$

## C EXPERIMENTAL DETAILS

This section serves to present additional details on the experiments that did not fit in the main paper, such as algorithm hyperparameters and additional details of the learning environments. These additional details will be provided for each experiment individually in the following subsections. The computations were executed on a desktop computer with a Geforce RTX 2080, 64 GB memory and an AMD Ryzen 9 3900X processor. All experiments were run in roughly a week of computation time on this single machine. The employed MushroomRL library is available under the MIT license.

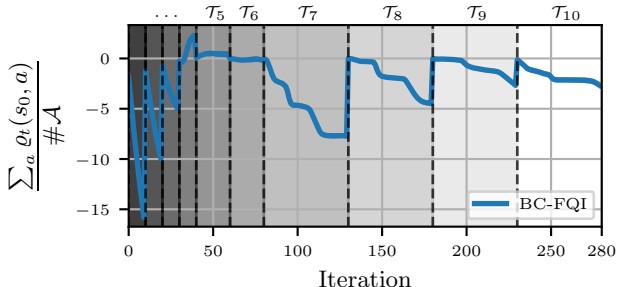

Figure 5: Average residual of the starting state $\mathbf{s}_0$ predicted by the learned $Q$-functions in the different learning tasks of the maze environment. The average and standard error (barely visible) is computed from 40 different seeds.

## C.1  CAR-ON-HILL

As mentioned in the main text, we used the FQI algorithm with extra-trees (Ernst et al., 2005) implemented in MushroomRL (D'Eramo et al., 2021) to conduct this experiment. The number of trees in the extra-trees implementation was set to 50, the minimum number of samples required to split an internal node to 5, and the minimum number of samples required to be a leaf node to 2. All other parameters were left to the implementation defaults of MushroomRL. Based on this implementation we created a boosted version that runs the original FQI algorithm on the residuals of the previously learned models. The hyperparameters for those extra-trees models were the same as for the regular FQI. The implementation can be found in the accompanying code. The car-on-hill environment used for this experiment is also implemented in MushroomRL.

## C.2  MAZE

The state-space of the maze environments consists of the two-dimensional coordinates $\mathbf{s}=[s_x, s_y]\in[0, 1]^2$. The initial state is $\mathbf{s}_0=[0, 0]$, and the goal state is $\mathbf{g}_0=[0.95, 0.95]$. The four actions displace the agent in $x$- or $y$-position by a value of 0.05. The discount factor is set to $\gamma=0.99$ and the horizon length is 200. We use a dense reward function $\mathcal{R}(\cdot, \cdot, \mathbf{s}') = -|\mathbf{g} - \mathbf{s}'|$ when $|\mathbf{g}-\mathbf{s}'| >= 0.1$, otherwise $\mathcal{R}(\cdot, \cdot, \mathbf{s}') = +5$ and the episode terminates. There are two walls centered at $x = 0.2$ and $x = 0.6$, both with width $w=0.05$, and height $h_t$ depending on the task $t$. We design a curriculum composed of 10 tasks, with $h_{1,...,9}=\{0.05, 0.15, ..., 0.85\}$ and $h_{10}=0.95$. Hence in the final task $\mathcal{T}_{10}$, there is only a very narrow passage through which the walls can be passed.
We again make use of the FQI algorithm with extra-trees implemented in MushroomRL to conduct this experiment, again using 50 trees, a minimum number of 5 samples to split an internal node, a minimum number of 2 samples to be a leaf node, and everything else left to the implementation defaults of MushroomRL. The main conceptual difference to the car-on-hill experiment is that exploration is guided by the currently estimated $Q$-function, as we are employing an $\varepsilon$-greedy exploration strategy to generate the samples for learning the $Q$-functions of the learning tasks. The exploration percentage $\varepsilon$ is annealed in later environments. More precisely, we set $\varepsilon_{1,...,10} = \{1., 1., 1., 1., .75, .75, .5, .5, .25, .25\}$ for the tasks $\mathcal{T}_1, ..., \mathcal{T}_{10}$.
Figure 5 visualizes the residuals of the starting state $\mathbf{s}_0$ that the $Q$-functions learned in the individual tasks, i.e., the additional values the $Q$-functions needed to predict in addition to the $Q$-value predicted from the ensemble of previously learned $Q$-functions. We see that the magnitude of residuals continuously decreases from task $\mathcal{T}_1$ to task $\mathcal{T}_6$. Switching to task $\mathcal{T}_7$ increases the magnitude of the residuals after which it then continuously declines again. The reason for this jump is that, while the optimal policy is largely unaffected by the presence of the small walls in the early learning tasks, a more significant detour is required starting from task $\mathcal{T}_7$.

## C.3  LINEAR SYSTEM CONTROL

The linear system that needs to be controlled is given by

$$\dot{\mathbf{s}} = \begin{bmatrix} \dot{x} \\ \ddot{x} \\ \dot{y} \\ \ddot{y} \end{bmatrix} = \begin{bmatrix} 0 & 1 & 0 & 0 \\ 0 & -0.1 & 0 & 0 \\ 0 & 0 & 0 & 1 \\ 0 & 0 & 0 & -0.1 \end{bmatrix} \begin{bmatrix} x \\ \dot{x} \\ y \\ \dot{y} \end{bmatrix} + \begin{bmatrix} 0 & 0 \\ 1 & 0 \\ 0 & 0 \\ 0 & 1 \end{bmatrix} \begin{bmatrix} a_x \\ a_y \end{bmatrix} = \mathbf{As} + \mathbf{Ba},$$

representing a simple, fully-actuated point-mass with a slight amount of friction. For the linear quadratic (LQ) task, to which the curriculum learning agents are exposed first, the reward function is given by

$$r(\mathbf{s}, \mathbf{a}, \mathbf{s}') = -\mathbf{s}^T \begin{bmatrix} 3 & 0 & 0 & 0 \\ 0 & 1 & 0 & 0 \\ 0 & 0 & 3 & 0 \\ 0 & 0 & 0 & 1 \end{bmatrix} \mathbf{s} - \mathbf{a}^T \begin{bmatrix} 0.1 & 0 \\ 0 & 0.1 \end{bmatrix} \mathbf{a}$$

and the initial state is given by $\mathbf{s}_0 = [-4 \ v_{x,0} \ 0 \ v_{y,0}]^T$, where $v_{x,0}$ and $v_{y,0}$ are sampled uniformly within $[-0.3, 0.3]$. In the target task, the agent receives the following reward signal

$$r(\mathbf{s}, \mathbf{a}, \mathbf{s}') = \begin{cases} -1, \text{ if } \|[x \ y]^T\|_2 \geq 0.2 \\ 0, \text{ else} \end{cases}$$

and its initial position is given by $\mathbf{s}_0 = [-4 \ v_{x,0} \ y \ v_{y,0}]^T$, where $v_{x,0}$ and $v_{y,0}$ are sampled uniformly within $[-0.1, 0.1]$ and $y$ within $[-0.4, 0.4]$. Furthermore, the agent needs to keep its $y$-position within $[-0.5, 0.5]$ in order to stay on the pathway indicated in Figure 4b and it faces an additional drift moving it away from the target, particularly

$$\dot{\mathbf{s}} = \mathbf{A}\mathbf{s} + \mathbf{B}\mathbf{a} + \begin{bmatrix} -0.5 \\ 0 \\ 0 \\ 0 \end{bmatrix}.$$

Consequently, the policies pre-trained in the LQ task do not quite reach the target but need to explore that they need to counteract this external force while staying on the pathway.

To learn with LSPI in the LQ task, we represent the policy by a PD-controller with explorative noise

$$\mathbf{a} \sim \mathcal{N}(\mathbf{K}\mathbf{s}, 0.1\mathbf{I})$$

as this ensures the quadratic form of the optimal $Q$-function mentioned in the main text. More precisely, it can be shown that

$$Q(\mathbf{s}, \mathbf{a}) = \boldsymbol{\theta}^T \boldsymbol{\phi}(\mathbf{s}, \mathbf{a}),$$

where $\boldsymbol{\phi}(\mathbf{s}, \mathbf{a})$ computes linear and quadratic terms of $\mathbf{s}$ and $\mathbf{a}$ (please see Bradtke (1993) for additional details). When learning with DQN in both the LQ and target task, the agent can choose between applying 8 discrete actions

$$\mathbf{a}_1 = \begin{bmatrix} 1 \\ 0 \end{bmatrix}, \ \mathbf{a}_2 = \begin{bmatrix} 0 \\ 1 \end{bmatrix}, \ \mathbf{a}_3 = \begin{bmatrix} -1 \\ 0 \end{bmatrix}, \ \mathbf{a}_4 = \begin{bmatrix} 0 \\ -1 \end{bmatrix},$$

$$\mathbf{a}_5 = \begin{bmatrix} .71 \\ .71 \end{bmatrix}, \ \mathbf{a}_6 = \begin{bmatrix} .71 \\ -.71 \end{bmatrix}, \ \mathbf{a}_7 = \begin{bmatrix} -.71 \\ .71 \end{bmatrix}, \ \mathbf{a}_8 = \begin{bmatrix} -.71 \\ -.71 \end{bmatrix}.$$

Consequently, we predict these discrete $Q$-values with the function approximator learned in the LQ task with LSPI when training on the target task. The $Q$-networks employed in DQN consist of two hidden layers of 128 neurons and ReLU activations. During training, we realized that the update frequency of the target network in the DQN algorithm had a strong impact on the final agent performance. Hence, we conducted a grid search testing values in $[25, 50, 100, 200, 500, 1000, 2000, 4000]$. For regular DQN on the target task, an update frequency of 200 performed best, while BC-DQN yielded the best results with an update frequency of 2000. For C-DQN, best results were achieved when using an update frequency of 1000 in the LQ task and an update frequency of 25 in the target task. When employing a shaping reward, taking 4000 environment steps between an update of the target network performed best for both boosted and non-boosted DQN. We used Adam for optimization of the network with a learning rate of $10^{-4}$ and $(\beta_1, \beta_2) = (0.9, 0.999)$. To exploit the previously learned policy in the target task for the curriculum methods, we used a fixed value of $\varepsilon = 0.1$ in the target task. For default DQN, we annealed $\varepsilon$ from 1 to 0.1 over the first 1000 environment steps.

As mentioned in the main paper, we additionally investigate residual policy learning (RPL) and probabilistic policy reuse (PPR) in this environment. Both approaches exploit a policy $\pi_{\text{source}}$ learned in a source task (for our experiment this is the LQ task) to accelerate learning in the target task. As the

methods are agnostic to the structure of the source task policy, we make use of the policy learned with LSPI, as this policy exhibits a more structured movement towards the target (see Figure 4b) and further requires fewer environment interactions to be learned (see Figure 4a). To implement RPL (Silver et al., 2018), we need to specify how to combine the actions $\mathbf{a}_i$ and $\mathbf{a}_j$ of the source task policy $\pi_{\text{source}}$ and the residual policy to be learned in the target task. In our experiment, we can simply add the actions $\mathbf{a}_i$ and $\mathbf{a}_j$ and then select the discrete action that is more similar to the combined action in terms of euclidean distance. If multiple actions are within the same minimum distance to the combined action, we choose the action that is more similar to the action of the residual policy, i.e. more similar to $\mathbf{a}_j$. This allows the residual policy to still move opposite to the previous policy if necessary while preserving the bias induced by the previous policy if the residual policy does not produce an action exactly opposing the one from the previous policy. RPL has an offset parameter that controls for how many steps the residual policy does not interfere with the source task policy. For the described way of combining actions, not interfering means simply choosing the same actions as the source task policy. We tried values of $0$, $2000$, and $4000$ steps for this offset value, where $2000$ performed best. PPR requires the user to choose a decay factor $\nu$ to anneal the likelihood of executing the source task policy via $p(\pi_{\text{source}}) = \nu^t$, where $t$ is the number of past interactions (note that we follow the interpretation of PPR by Brys et al. (2015) for transfer from a single source task). We tried three different values $\nu=0.998$, $\nu=0.9996$ and $\nu=0.99988$, with $\nu=0.998$ performing best. With this value, $\nu^t=0.1$ after $t \approx 1150$ environment steps. As for the other baselines, we do a grid search over the target network update frequency, choosing the best. For RPL, a value of $25$ performed best. For PPR, a value of $200$ performed slightly better than lower ones. For both methods, update frequencies larger than $1000$ did not learn at all on the target task.

