# OpenReview forum: "Boosted Curriculum Reinforcement Learning"
_ICLR.cc/2022/Conference — ICLR 2022 Poster_

### Official Review · Reviewer_YXe6 · 2021-10-25

**Correctness:** 3
**Technical Novelty And Significance:** 3
**Empirical Novelty And Significance:** 3
**Recommendation:** 6
**Confidence:** 4

**Main Review:**

Strengths:
The boosting method is an interesting and promising way to adapt the change of task difficulty. As far as I know, this is the first paper that applies boosting to the RL curriculum learning. The experiment is solid and results are impressive.

Weakness:
My concern is on the theoretical analysis. I found the formulation and analysis are poorly written. Many notations are not clear. The proof is either trivial or a direct application of previous results.

Here are the detailed concerns.

1. Can authors discuss how strong the assumption on the operator $S$ in Theorem 2 is? It seems a very strong assumption since it holds uniformly for all $y \in \mathcal{B}(\mathcal{S} \times \mathcal{A})$.

2. It is weird that the author states that the approximation error comes with two aspects and the second one depends on the nonlinear operator $S$, while the definition of approximation error in Equation (1) is independent of $S$.

3. I think Theorem 1 is too complicated to interpret. For example, what is the behavior of the R.H.S in Theorem 1? The authors should elaborate more on the intuition of Theorem 1 (discussing the meaning of each term).  If the details of Theorem 1 is not important, the author may present only an informal version of Theorem 1 that highlights the high-level idea.

4. In the definition of the nonlinear operator $S$, it is not clear what is $\mu$-norm. If I am correct, $\mu$ is the probability measure over $\mathcal{S} \times \mathcal{A}$ and the empirical version $\hat S$ is the empirical error evaluated on the dataset $\mathcal{D}_t$. It is also not clear what $\mu$ means in Algorithm 1.

5. The notation is conflicting at the start of Section 4.1. It seems $Q_t^0$ represents the starting of task $t$. That means when task changes, $k$ should be reset to 0. However, the author also writes that $Q_{t+1}^{k+1} \approx T_{t+1}^{*} Q_{t}^{k}$.

6. I am assuming that Theorem 2 is analyzing the $Q_t^k$ using the accurate $S$ instead of the empirical version. The authors should make it clear about this point.

7. It seems there are two versions of $Q_{t}^k$ in the paper, one generated from the optimal Bellman operator and the other one from the empirical one. The author should use different notations for them. For example, in Equation (8), the approximation error should be using the empirical one if I understand it correctly.

8. Essentially, Theorem 4 concerns only the error of the last task. There should be more discussions on $\epsilon_i^0$, which is also some approximation using finite samples. I think a more interesting direction is that when the complexity of the task is actually increasing, when the algorithm without boosting will suffer a high approximation error if they only use simple models. If they use complex models, then the VC-dimension could be high even for the simpler tasks, which slows down the training.

**Summary Of The Paper:**

This paper proposed a curriculum value-based reinforcement learning algorithm. Inspired by boosting methods, the algorithm learns on the Bellman residuals between the value function of the current task and that of the previous task. This paper provides analysis on the convergence of the algorithm and some finite-sample guarantee on the output value function. The method is tested on four different problems and is compared with two 3 baseline methods.

**Summary Of The Review:**

I recommend this is a borderline paper due to the issues I raised on the theoretical part and the notations.

---

> ### Author Response · Authors · 2021-11-18
> **Answer to Reviewer YXe6 - 1**
>
> We thank the reviewer for the effort put into reading our paper and the detailed comments. In the following, we aim at addressing all the concerns.
>
> >Can authors discuss how strong the assumption on the operator $S$ in Theorem 2 is? It seems a very strong assumption since it holds uniformly for all $y \in \mathcal{B}(\mathcal{S} \times \mathcal{A})$.
>
> Our assumption implies that the approximation power of the used parameterized class of function is good enough to represent the target function, stating that the approximation error $\lVert(I-S)y\rVert_\mu$ has an upper bound that is proportional to the magnitude of the function $\lambda\lVert y\rVert_\mu$ for all $y \in \mathcal{B}(\mathcal{S}\times\mathcal{A})$. This assumption holds for many reasonable choices of function approximators, and it is used in related works [BFQI], and in the form of even tighter bounds in works deriving finite-time bounds for policy and value iteration [EAPI, EAVI] and Supervised Learning literature [BOOST]. We added comments to Theorem 2 to better clarify this point.
>
> >It is weird that the author states that the approximation error comes with two aspects and the second one depends on the nonlinear operator $S$, while the definition of approximation error in Equation (1) is independent of $S$.
>
> The approximation error $\varepsilon_k$ is dependent from $S$ because $Q^{k+1}$ is the approximation of the $Q$-function obtained after the application of the projection $S$ to the Bellman operator $T^*Q^k$. We improved the description of Equation (1) to highlight this.
>
> >I think Theorem 1 is too complicated to interpret. For example, what is the behavior of the R.H.S in Theorem 1? The authors should elaborate more on the intuition of Theorem 1 (discussing the meaning of each term). If the details of Theorem 1 is not important, the author may present only an informal version of Theorem 1 that highlights the high-level idea.
>
> We agree with the reviewer that the bound is showing complexities that go beyond the scope of our work, in particular the definition of the concentrability coefficients $C_{\text{VI},\rho,\mu}$ is superfluous. We removed the definition of concentrability coefficients referring to the original paper [REGRL]. The high-level idea of Theorem 1 is that the upper bound of the magnitude of the difference between the optimal $Q$-function $Q^*$ and the estimate at the K-th iteration $Q^K$ is proportional to the magnitude of the approximation errors $\varepsilon_k$. We added this statement in the revised paper.
>
> >In the definition of the nonlinear operator $S$, it is not clear what is $\mu$-norm. If I am correct, $\mu$ is the probability measure over $\mathcal{S}\times\mathcal{A}$ and the empirical version $\hat{S}$ is the empirical error evaluated on the dataset $\mathcal{D}_t$. It is also not clear what $\mu$ means in Algorithm 1.
>
> Thank you for pointing this out. For the sake of clarity, we recall that the $\mu$-norm is the $L_p$-norm w.r.t. a probability measure $\mu$. As stated in Section 3, the omitted $p$ implies that we consider an $L_2$-norm. As correctly stated by the reviewer, the empirical nonlinear operator $\hat{S}$ is computed using the dataset $\mathcal{D}_t$. Therefore, Algorithm 1 should use $\hat{S}$ instead of $S$. We fixed this in the revised paper.
>
> >The notation is conflicting at the start of Section 4.1. It seems $Q_t^0$ represents starting of task $t$. That means when task changes, $k$ should be reset to $0$. However, the author also writes that $Q_{t+1}^{k+1}\approx T^*_{t+1}Q^k_t$.
>
> Thanks for spotting this error. We fixed it in the revised paper by removing  $Q_{t+1}^{k+1}\approx T^*_{t+1}Q^k_t$, that was indeed wrong, and superfluous.
>
> [EAVI] Munos, Rémi. "Error bounds for approximate value iteration." Proceedings of the National Conference on Artificial Intelligence. Vol. 20. No. 2. Menlo Park, CA; Cambridge, MA; London; AAAI Press; MIT Press; 1999, 2005.
>
> [EAPI] Munos, Rémi. "Error bounds for approximate policy iteration." International Conference on Machine Learning. Vol. 3. 2003.
>
> [BFQI] Tosatto, Samuele, et al. "Boosted fitted q-iteration." International Conference on Machine Learning. PMLR, 2017.
>
> [BOOST] Bühlmann, Peter, and Torsten Hothorn. "Boosting algorithms: Regularization, prediction and model fitting." Statistical science 22.4 (2007): 477-505.
>
> [REGRL] Farahmand, Amir-massoud. "Regularization in reinforcement learning." (2011).

---

> > ### Author Response · Authors · 2021-11-18
> > **Answer to Reviewer YXe6 - 2**
> >
> > >I am assuming that Theorem 2 is analyzing the $Q^k_t$ using the accurate $S$ instead of the empirical version. The authors should make it clear about this point.
> >
> > We addressed this concern by answering to the point below.
> >
> > >It seems there are two versions of $Q^k_t$ in the paper, one generated from the optimal Bellman operator and the other one from the empirical one. The author should use different notations for them. For example, in Equation (8), the approximation error should be using the empirical one if I understand it correctly.
> >
> > Thanks for pointing this out. We understand that this can cause confusion. However, we believe it is better to not complicate the notation and equally denote the $Q$-function obtained with the empirical Bellman operator and the one obtained with the optimal Bellman operator. We added a comment in the revised paper to clarify this important detail. We hope this is enough. We will be willing to reconsider our choice in case the reviewer will still be concerned about it.
> >
> > >Essentially, Theorem 4 concerns only the error of the last task. There should be more discussions on $\varepsilon^0_i$, which is also some approximation using finite samples. I think a more interesting direction is that when the complexity of the task is actually increasing, when the algorithm without boosting will suffer a high approximation error if they only use simple models. If they use complex models, then the VC-dimension could be high even for the simpler tasks, which slows down the training.
> >
> > Thank you for this insightful comment. We want to clarify that the result provided by Theorem 4 applies to every task $t$ in the curriculum, not only on the target task.
> > We agree on the benefit of increasing the model complexity along with the task complexity in the boosted framework. Indeed, the linear system control experiment aims to highlight this benefit. In this experiment BC-DQN (i.e. the boosted curriculum), pre-trains in an LQ problem in which we can represent the $Q$-function using simple linear models with quadratic features. Only for the target task, we use a neural network to represent the $Q$-function. The results show that the use of initially simple models allows for faster learning.

---

> > > ### Comment · Reviewer_YXe6 · 2021-11-18
> > > **Thanks for the response**
> > >
> > > Thanks for the response. I am satisfied with the improvement.
> > >
> > > There are a typo I see: In Theorem 1, $B(\mathcal{X} \times \mathcal{A}, Q_{max})$ should be $\mathcal{B}(\mathcal{X} \times \mathcal{A}, Q_{max})$.
> > >
> > > > We agree with the reviewer that the bound is showing complexities that go beyond the scope of our work, in particular the definition of the concentrability coefficients
> > >
> > > I think it is better to write an informal version of Theorem 1 to just give the high-level idea instead of introducing $C_{VI}$ and $\alpha_i$ without defining them.
> > >
> > > > Thank you for this insightful comment. We want to clarify that the result provided by Theorem 4 applies to every task $t$ in the curriculum, not only on the target task.
> > >
> > > I see. So one can apply Equation (11) iteratively to have a bound on $\epsilon_t^k$, without the dependence on $\epsilon_i^0$. The current version is less meaningful because when we look at $\epsilon_t^k$, we don't know how large are $\epsilon_i^0$.

---

> > > > ### Author Response · Authors · 2021-11-19
> > > > **Answer to Reviewer YXe6**
> > > >
> > > > We thank the reviewer for the positive feedback about the revised paper.
> > > >
> > > > In the following, we address the remaining concerns.
> > > >
> > > > >There are a typo I see: In Theorem 1, $B(\mathcal{X}\times\mathcal{A},Q_{\text{max}})$ should be $\mathcal{B}(\mathcal{X}\times\mathcal{A},Q_{\text{max}})$.
> > > >
> > > > Thanks for spotting this typo. Actually, also $\mathcal{X}$ should be replaced with $\mathcal{S}$, following the notation used in our paper. We fixed this in the revised paper.
> > > >
> > > > >I think it is better to write an informal version of Theorem 1 to just give the high-level idea instead of introducing $C_{\text{VI}}$ and $\alpha_i$ without defining them.
> > > >
> > > > Thank you for this comment. The revised paper contains now a high-level version of Theorem 1 and an additional explanation. For completeness, we kept the extended version of the theorem moving it in the appendix. We still refer to Farahamand, 2011 for the definition of the concentrability coefficients and $\alpha_i$, since they would require additional definitions and equations that we consider excessive to report in this work. We hope this resolves the reviewer's concern.
> > > >
> > > > >I see. So one can apply Equation (11) iteratively to have a bound on $\varepsilon_t^k$ without the dependence on $\varepsilon_i^0$. The current version is less meaningful because when we look at $\varepsilon_t^k$, we don't know how large are $\varepsilon_i^0$.
> > > >
> > > > We are unsure about what the reviewer means. Equation (11) applies to the approximation error $\varepsilon^k_t$ for each task $t$ at any timestep $k$. The second component of the R.H.S. of the equation expresses the dependency of the approximation error $\varepsilon^k_t$ on the propagation of previous approximation errors $\varepsilon^0_i$, i.e., for each already visited task $\mathcal{T}_i$. We would be glad to resolve this concern if the reviewer could clarify it.

---

> > > > > ### Comment · Reviewer_YXe6 · 2021-11-21
> > > > > **Clarify**
> > > > >
> > > > > Thanks for the response. Let me clarify the concerns on Equation (11). I am wondering how large propagation error is compared with e.g. approximation error and estimation error since $\epsilon_i^0, I = 1, \dots, t-1$ are unknown. However, one can get rid of the propagation error term by plugging in $\epsilon_i^0$ using equation (11) iteratively (for $\epsilon_1^k$ there is no propagation error, then one can plug it into $\epsilon_2^k$ and so on). I think that form will be more meaningful to read.

---

> > > > > > ### Author Response · Authors · 2021-11-22
> > > > > > **Thanks for the clarification**
> > > > > >
> > > > > > Thanks for the clarification. We understand that we could get rid of the propagation error term by applying Equation 11 recursively, as the reviewer says. As correctly stated, since for the bound of $\varepsilon^k_1$ there is no propagation error, we would be able to reformulate the equation only in terms of approximation and estimation error. However, we think that this extended version of the formula would make the equation more difficult to read, without being significantly more informative.  We think that the current version, also used in previous works [VPI, BOOST], is well-suited for evincing the dependence on previous approximation errors in a compact way, while not preventing to correctly interpreting the meaning of the theorem and the possibility of deriving the extended formula mentioned by the reviewer.
> > > > > >
> > > > > > [VPI] Farahmand, Amir Massoud, and Doina Precup. "Value Pursuit Iteration." NIPS. 2012.
> > > > > >
> > > > > > [BOOST] Tosatto, Samuele, et al. "Boosted fitted q-iteration." International Conference on Machine Learning. PMLR, 2017.

---

### Official Review · Reviewer_N1P2 · 2021-10-30

**Correctness:** 3
**Technical Novelty And Significance:** 3
**Empirical Novelty And Significance:** 2
**Recommendation:** 6
**Confidence:** 3

**Main Review:**

The authors propose a Curriculum Learning method to "learn" a residual for each task, instead of trying to learn from scratch each of them.
The proposed method is interesting and creative, and seems to be beneficial to the learning process in the empirical evaluation.

My main concern is the omission of the many related works. Although the authors evaluate the algorithm only in simple environments (which means it's relatively easy to implement other baseline methods), they do not compare against the results of any related methods from [1] or [2], for example (the former is.cited in the manuscript).

[1] Sanmit Narvekar, Bei Peng, Matteo Leonetti, Jivko Sinapov, Matthew E Taylor, and Peter Stone. Curriculum learning for reinforcement learning domains: A framework and survey. Journal of Machine Learning Research, 21(181):1–50, 2020.

[2] Silva, Felipe Leno, and Anna Helena Reali Costa. "A survey on transfer learning for multiagent reinforcement learning systems." Journal of Artificial Intelligence Research 64 (2019): 645-703.

Furthermore, some sentences show lack of familiarity with the literature, for example: "[...]contrary to curriculum RL which uses the same functional space " (not all Curriculum Learning methods do that). Therefore, I would really like to see a performance comparison between the proposed method and state-of-the-art Curriculum Methods, instead of just against "baselines".


----
minor
----

Shouldn't it be "C-FQI" in the legend of Figure 1?





-----
Post-rebuttal
-----
The authors tried to address some of my concerns, in special adding some text in an appendix commenting on the distinction of their paper from TL. While I appreciate the (necessary) effort, I don't see the reason for adding this as an appendix, and think it would have been better to incorporate it to the main text even if it would require more effort to make sure space limitations are not exceeded. My evaluation was already positive, and the author response and other reviewers' evaluation had nothing that would make me lower my grades. Therefore, my evaluation is maintained .

**Summary Of The Paper:**

The paper proposes a new curriculum for RL method. The basic idea is to "learn" a residual for each task, modeling in this way how the tasks differ. The research topic is highly relevant and contemporary and the theoretical analysis of the method is interesting. The major drawback is the empirical evaluation.

**Summary Of The Review:**

Paper investigating a relevant research topic. The manuscript contributes interesting ideas and the proposed method seems to have good performance. However, the lack of comparison against state-of-the-art methods makes it hard to assess how good the method is compared to other related works.

---

> ### Author Response · Authors · 2021-11-11
> **Answer to Reviewer N1P2**
>
> We thank the reviewer for the provided comments on our work. We aim to address the raised question/concerns in the following:
>
> >My main concern is the omission of the many related works. Although the authors evaluate the algorithm only in simple environments (which means it's relatively easy to implement other baseline methods), they do not compare against the results of any related methods from [1] or [2], for example (the former is.cited in the manuscript).
>
> We agree that the perceived lack of baselines may seem confusing to readers and we thank the reviewer for pointing this out. We remark that the main purpose of our experimental results is to substantiate the claims provided in our theoretical analysis. For this reason, we focused on an ablation study of boosting-related techniques. Although carefully considering related works in literature, we have not been able to find adequate baselines, and we hope to provide a satisfying explanation for this in the following. Our method is not investigating ways to generate curricula by creating or sequencing tasks (as e.g. done by GoalGAN or ALP-GMM), but it is situated in what [CRL] refers to as a value function transfer setting, in which the value function learned in the current task is used to initialize the value function estimate of the next task in the curriculum. In this scenario, we could only find the work on Progressive Neural Networks (PNNs) [ProgNN] that differs from the previously outlined default approach to value function transfer. However, PNNs represent a particular architectural design decision and could indeed be incorporated in our boosting framework. Consequently, we did not compare to this method as it does not present an alternative approach to value function transfer. If the reviewer has a particular suggestion for a well-suited baseline in the value transfer setting, we are definitely willing to include it in the revised paper.
> In the meanwhile, to avoid this confusion in the future, we slightly revised the related work section to better situate our method in the broad field of curriculum RL.
>
> [CRL] Narvekar, Sanmit, et al. "Curriculum learning for reinforcement learning domains: A framework and survey." arXiv preprint arXiv:2003.04960 (2020).
>
> [ProgNN] Rusu, Andrei A., et al. "Progressive neural networks." arXiv preprint arXiv:1606.04671 (2016).
>
> > Furthermore, some sentences show lack of familiarity with the literature, for example: "[...]contrary to curriculum RL which uses the same functional space " (not all Curriculum Learning methods do that)
>
> Thank you for drawing our attention to the cited sentence. We agree that the sentence is too generic and we fixed it in the paper. In case there are remaining sentences that could be improved, we would be glad if the reviewer could mention them.
>
> > Shouldn't it be "C-FQI" in the legend of Figure 1?
>
> We double-checked the legend of Figure 1 but could not find any problems. Can the reviewer explain her/his confusion about the legend?

---

> > ### Comment · Reviewer_N1P2 · 2021-11-12
> > **further discussion**
> >
> > > We remark that the main purpose of our experimental results is to substantiate the claims provided in our theoretical analysis.
> >
> > I understand that, but given that ICLR is one of the most competitive conferences in the subject, I would expect that the all papers submitted here either perform comparison against the state-of-the-art approaches or at least provide very explicit and sufficient explanation on why it was not possible to perform this comparison.
> >
> > > but it is situated in what [CRL] refers to as a value function transfer setting
> >
> > Virtually all of the approaches cited on the Survey I mentioned in my review focus specifically on value transfer (unsurprisingly, given that the survey is focused on transfer learning). While I expect that some assumptions or particularities in methods could possibly render them very hard to reimplement in another domain and thus to compare against your approach, I would expect at least to reason for not including the paper to be included in the manuscript, for ones published after ~2016 .
> >
> > > In case there are remaining sentences that could be improved
> >
> > I can recall that (Silva & Costa, 2018), from the survey I mentioned in my review, specifically transfers across tasks with different state-action spaces. I am pretty sure a number of other approaches mentioned in both surveys also do that, as I have encountered a good number of approaches not restricted to the same state-action space in the last years (in general, approaches that transfer knowledge from simpler versions of the task to bigger ones are very very common).

---

> > > ### Author Response · Authors · 2021-11-18
> > > **Further Discussion**
> > >
> > > We'd like to thank reviewer N1P2 for the quick additional feedback based on our first answer. We now additionally evaluate residual policy learning [RPL] and probabilistic policy reuse [PPR] in the linear control task of Section 6.3. We chose these two methods, as they both, like boosting, do not require additional effort, e.g. in the form of transition models or learned shaping rewards, to facilitate the transfer between tasks. As can be seen in the updated Figure 4, these methods improve over a regular curriculum (in the case of PPR only marginally), however, are only on par with or worse than regular learning in terms of final agent performance.
> > >
> > > [RPL] Silver, Tom, et al. "Residual policy learning." (2018).
> > >
> > > [PPR] Fernández, Fernando, and Manuela Veloso. "Probabilistic policy reuse in a reinforcement learning agent." AAMAS. 2006.

---

### Official Review · Reviewer_hq3v · 2021-11-03

**Correctness:** 4
**Technical Novelty And Significance:** 3
**Empirical Novelty And Significance:** 3
**Recommendation:** 8
**Confidence:** 4

**Main Review:**

**Pros**
* Writing and Presentation: The paper was clearly written and the graphs were well annotated.
* BCRL Framework: The BCRL framework was interesting (in particular the idea of boosting using residuals). Some of the presented theorems were a bit mathematically involved, but the presentation of the ideas systematically helped parse the key ideas in the proof.

**Cons**
* Experiments and evaluation: While the setup is largely well explained, it would have been clear if at least one paragraph explained the details of the Linear System Control environment. From the main paper, it was not clear what was meant by the LSPI applied to LQR -- some of the details from the appendix can be brought to the main paper. Furthermore, the reasoning given for the decline in the performance of CRL in the environment needs more clarification/insights. I was not entirely convinced by the reasoning provided (on unlearning of Q-function when the task was switched to the target task).

* Clarification question: Am I correct in understanding that, the proof (beginning with Proposition 1) assumes that convergence to the optimal state-action function will take place in a finite number of iterations? Extending the ideas presented, can it be said that the subsequent approximation error of the residuals becomes smaller as the number of tasks in the curriculum increase, given that for a task the error of BCRL is bounded?

* Clarification question: The terms in Eq (11) can be made more intuitive in the paragraph explaining them. This was slightly difficult to parse.

* The limitations of the approach could have been addressed better.

**Additional Comments**:

Can the presented methodology be extended to account for the increasing task complexity in the curriculum and weigh the importance of each task gradually presented in learning the state-action value?

**General Summary**:

Overall I thought the paper presented the key ideas in a clean way. The mathematical framework is very interested and opens avenues for future research in the area -- both practical and theoretical. The experimental setup for the Linear Control System could have been better explained.

*Originality*:  Moderate

*Clarity*: Good

*Quality*: Good

*Significance*: Moderate to High

**Summary Of The Paper:**

The paper tackles the problem of curriculum learning and presents a new technique (boosted CRL) that provides a tighter bound on the approximation error of the action-value function than standard curriculum learning. The authors show both theoretically and empirically the effectiveness of the approach over standard CRL. The presented methodology learns a function approximation based on the sum of `residuals'. The evaluation was done on 3 environments that had scope for the generation of a curriculum by varying the reward and exploration factors -- car on hill, maze, linear system control. Results show that BCRL performs significantly better than CRL in all environments.

**Summary Of The Review:**

Overall I liked the ideas presented in the paper and think it provides a good starting point for analyzing theoretically, the non-trivial task of curriculum learning in RL. The idea of using the sum of residuals to model the state-action function approximation was interesting. I am inclined to accept the paper because of its promise and utility in inspiring similar approaches to curriculum learning (which provide a framework for theoretical analysis), though the experimental set-up could have been made clearer and limitations of the approach be made explicit.

---

> ### Author Response · Authors · 2021-11-18
> **Answer to Reviewer hq3v**
>
> We thank the reviewer for the insightful comments and positive feedback, allowing us to further improve the paper. In the following, we answer to the raised questions/concerns.
>
> >From the main paper, it was not clear what was meant by the LSPI applied to LQR -- some of the details from the appendix can be brought to the main paper. Furthermore, the reasoning given for the decline in the performance of CRL in the environment needs more clarification/insights. I was not entirely convinced by the reasoning provided (on unlearning of Q-function when the task was switched to the target task).
>
> We revised this section aiming to better explain the idea behind the application of LSPI to the LQ system. We additionally added a reference to the relevant appendix. We further reformulated the final hypothesis about the ‘’unlearning’’ of the stabilizing behavior, as it is hard to convey in a plot. However, the clear benefit visualized in Figure 4b) is that the learned LSPI policy induces a more robust bias towards the goal position that is preserved throughout learning, as the Q-function learned with LSPI is fixed in the boosted setting.
>
> >Am I correct in understanding that, the proof (beginning with Proposition 1) assumes that convergence to the optimal state-action function will take place in a finite number of iterations? Extending the ideas presented, can it be said that the subsequent approximation error of the residuals becomes smaller as the number of tasks in the curriculum increase, given that for a task the error of BCRL is bounded?
>
> The convergence to the optimal state-action value function is generally not guaranteed in the Approximate Value Iteration setting. However, the asymptotic behaviour can be analyzed, and we know that a sufficiently good estimate is reached in a finite number of steps [FQI]. As we also state in the paper and show empirically in Figure 5 in the appendix, the residuals become smoother functions closer to zero for increasingly complex tasks. This makes the residuals progressively easier to learn, resulting in smaller estimation errors as stated in Theorem 4.
>
> >The terms in Eq (11) can be made more intuitive in the paragraph explaining them. This was slightly difficult to parse.
>
> We totally understand that the Equation 11 is not immediately easy to parse. However, we are unsure about how to improve the visualization or the description. We remark that Theorem 4 is mostly based on Theorem 11.5 of [NPAR], that we use for deriving our result, from which it inherits some complexities. We try to clarify the components of the equation here. Some of the components used in the bound provided in Theorem 4 are already used in other equations. The residual $\bar{\varrho}$, used in Theorem 3, is the regular residual in the curriculum AVI setting, i.e., the residual obtained without using our boosting procedure. $\varepsilon$ is the approximation error as defined in Equation 8. $L$ is the Lipschitz coefficient of the optimal Bellman operator used in Theorem 2.
> The components $B^k_t$ and $V_{\mathcal{F}_+}$ are introduced in Theorem 4, and come from Theorem 11.5 of [NPAR]. The detailed motivation for the presence of these two components is rooted in the complex theoretical framework for function regression presented in [NPAR], that we consider beyond the scope of this work. Nevertheless, we want to point out that the presence of a VC-dimension in the bound is not surprising, as the approximation error bound is reasonably related to the expressive power of the used function space $\mathcal{F}$.
> We hope this addresses the concern of the reviewer. If not, we will greatly appreciate any suggestions on how to improve the description of the theorem.
>
> >The limitations of the approach could have been addressed better.
>
> We added a short paragraph in the conclusion section discussing current limitations/unaddressed issues, such as how to combine the method with automatic curriculum generation methods and how to avoid the linear increase of the network size w.r.t. the number of tasks.
>
> >Can the presented methodology be extended to account for the increasing task complexity in the curriculum and weigh the importance of each task gradually presented in learning the state-action value?
>
> Conceptually, the boosting framework allows to scale the learned Q-function, allowing to gradually decay the influence of the Q-functions learned in earlier tasks. However, we believe that this would require additional theoretical investigations regarding potentially arising biases and how to overcome them.
>
> [CRL] Narvekar, Sanmit, et al. "Curriculum learning for reinforcement learning domains: A framework and survey." Journal of Machine Learning Research (2020).
>
> [FQI] Ernst, Damien, Pierre Geurts, and Louis Wehenkel. "Tree-based batch mode reinforcement learning." Journal of Machine Learning Research (2005).
>
> [NPAR] Györfi, László, et al. A distribution-free theory of nonparametric regression. Vol. 1. New York: Springer, 2002.

---

> > ### Comment · Reviewer_hq3v · 2021-11-20
> > **Thank you for the Clarifications**
> >
> > Thanks for the clarifications. The references were helpful in parsing the Theorems! The additional explanations helped clarify the ideas of Theorem 2. I understand that explaining Eq 11 is not easy. A starter suggestion could be to perhaps, use a color-coded annotation of the terms in the equation which can explicitly point out the 3 important components of the bound (as described later in the paragraph): (i) the approximation error of the residual, (ii) the propagation error due to approximation errors at previous iterations, (iii) the estimation error due to the regression problem with a finite amount of samples.

---

> > > ### Author Response · Authors · 2021-11-21
> > > **Added brace annotation to Equation 11**
> > >
> > > Thank you for the suggestion. In the revised paper, we decided to add a brace annotation to explicitly point out the 3 important components of the bound. We hope this enhances the readability and interpretability of the theorem.

---

> > > > ### Comment · Reviewer_hq3v · 2021-12-01
> > > > **Thanks for the revisions!**
> > > >
> > > > Thanks for the revisions and updates! LSC (Section 6.3) is clear now. After following the discussions, and the revisions, I will keep my score.

---

### Official Review · Reviewer_TFaS · 2021-11-17

**Correctness:** 3
**Technical Novelty And Significance:** 3
**Empirical Novelty And Significance:** 3
**Recommendation:** 6
**Confidence:** 4

**Main Review:**

The paper makes an interesting progress/contribution towards applying reinforcement learning methods for real-world challenges (hard exploration problems). The proposed algorithm is novel and justified both theoretically and empirically.

I haven’t carefully checked the theoretical analysis (Section 5) and the corresponding proofs.

The paper is overall well written and easy to follow. However, the related work discussion seems limited: missing references for the works from (i) curriculum design via environment design, and  (ii) knowledge transfer via reward shaping literature.

----

In the introduction, the authors state that the existing (value-based) curriculum RL methods are constrained to use the same function approximator throughout all the tasks; this is because the trained Q-value function of the current task in the sequence is used as the initialization for the next task. However, as in [Brys et al. 2015], one can utilize potential-based reward shaping techniques (a principled approach) to transfer knowledge between tasks. In this case: (i) the next task in the sequence will receive a dense/informative reward function that will ease the learning while maintaining the policy invariance property, and (ii) also, has the flexibility of choosing different functions approximators for each task in the sequence. Thus, it is important to both conceptually and empirically compare your proposed curriculum method with the curriculum methods with knowledge transfer based on reward shaping methods (e.g., [Brys et al. 2015]).

[Brys et al. 2015] Brys et al. Policy Transfer using Reward Shaping. AAMAS, 2015.

----

I had a few confusions related to the experiments (Section 6). Please clarify them.

I am a bit confused about the methods compared (mainly FQI and B-FQI) in the experiments section:
(i) BC-FQI is a method clear: FQI blended with Algorithm 1 (boosting + curriculum)
(ii) C-FQI: FQI trained over the curricula of tasks; here, the knowledge transferred between tasks is the trained Q-value function (as in existing value-based curriculum RL methods). Is this right?
(iii) FQI: trained only on the data from the target task or random mixture of data from all the tasks? The FQI (red color) curve in Figure 1 only corresponds to convergence/performance of FQI on the target task T3; or it is region-wise segmented for T1, T2, T3?
(iv) B-FQI: same confusion as in FQI. If it is trained on the target task only, what is the boosting component here?

A clear description of each method (or even pseudocode in the appendix) compared would be helpful for the reader.

In summary,
1/ What is the difference between the curriculum methods? Are the non-curriculum methods only trained on the target task or they are also trained on the same pool of tasks as curriculum methods but presented in a random ordering?

2/ If the non-curriculum methods are trained only on the single target task, what is the role of boosting here, i.e., how is the sum of residuals of different tasks applicable here?

3/ “The boosted methods that do not use a curriculum will introduce the additional approximators at the same iteration at which the curricula switches between tasks.” -- this statement is not clear to me. Please clarify.

----

A question/suggestion/comment regarding the Maze experiment:
“It obtains a dense reward based on the distance to the target ...”
Isn’t a dense reward already ease the learning process of the RL agent? Of course, here, the size of the two walls will alter the hardness of the environment (by changing the transition dynamics). Still, considering the target task with a sparse reward that obtains a reward only when it reaches the goal would make it a complex problem to solve. Have you tested the sequence of tasks with such sparse rewards?

The hardness of the sequence of tasks can be varied by: (i) fixing the reward function, and varying the transition dynamics of each task, (ii) fixing the transition dynamics, and varying the reward function of each task, and/or (iii) varying both reward and dynamics. I see that both Car-on-Hill and Maze experiments fall under case (i). Have you tested the case (ii)?

A simple experiment to cover case(ii): Consider a (stochastic) chain with length L (very large), and the goal is at the end of the chain. Here (keeping L fixed), one can vary the hardness of the tasks by defining reward functions with different granularity: sparse goal-oriented reward (target task), landmark-based rewards (intermediate tasks), and optimal myopic rewards obtained via potential-based reward shaping (easiest task).

To cover case(i) with the chain example, fix the reward as a sparse goal-oriented reward, and gradually increase the length of the chain.


**Summary Of The Paper:**

This paper studies a fundamental issue in curriculum design for a complex reinforcement learning problem (target task). When the target task is complex to solve with direct training on it, the learner can be trained over a sequence of tasks with increasing difficulty. Here, the tasks in the sequence vary in either the transition dynamics or reward function. The central question of the paper: developing a principled method for transferring the knowledge (Q-value function or residuals) gained in the current task of the curricula to the next task. The authors propose to use the sum of residuals trained on each task for knowledge transfer and develop a novel curriculum RL method called Boosted Curriculum Reinforcement learning. By leveraging techniques from approximate value iteration literature, they theoretically justify their choice for knowledge transfer and their algorithmic proposal. The authors also conduct interesting empirical investigations on fleshing out the importance of boosting and the curriculum components of their algorithm.

**Summary Of The Review:**

The paper proposes an interesting and novel algorithm for curriculum RL. The paper is overall well written. However, the conceptual and empirical comparison to knowledge transfer via reward shaping methods is missing. There were some clarity issues in the experiments. If these two issues are addressed, I am marginally inclined towards acceptance.

---

> ### Author Response · Authors · 2021-11-18
> **Answer to Reviewer TFaS**
>
> We thank the reviewer for the feedback and the suggestions for additional experiments. We now answer to the raised questions and discuss the suggestions regarding baselines and experiments.
>
> > Thus, it is important to both conceptually and empirically compare your proposed curriculum method with the curriculum methods with knowledge transfer based on reward shaping methods (e.g., [Brys et al. 2015]).
>
> We are thankful for the additionally suggested reference and added it to our discussion of transfer methods in appendix A of the updated paper. We, however, chose to evaluate residual policy learning [RPL] and probabilistic policy reuse [PPR] in section 6.3 due to two reasons:
> 1. RPL and PPR do not introduce additional complexities such as the method of Brys et al, which needs to learn the shaping function via an additional reinforcement learning objective. We believe that due to the simplicity of boosted curricula (which does not introduce any additional hyperparameters that need to be chosen over non-boosted curricula), the evaluated baselines should not introduce major additional complexities as well.
> 2. Brys et al. contrasted their method to PPR (as well as a combination of their method and PPR). Although their method sometimes outperformed PPR, PPR showed strong gains compared to regular learning, sometimes performing similar to the method of Brys et al.
>
> We hope that the reviewer understands this choice and that these additional evaluations address the concerns regarding baselines.
>
> [Brys et al. 2015] Brys et al. Policy Transfer using Reward Shaping. AAMAS, 2015.
>
> [RPL] Silver, Tom, et al. "Residual policy learning." (2018).
>
> [PPR] Fernández, Fernando, Javier García, and Manuela Veloso. "Probabilistic policy reuse for inter-task transfer learning." Robotics and Autonomous Systems (2010)
>
> > In summary, 1/ What is the difference between the curriculum methods? Are the non-curriculum methods only trained on the target task or they are also trained on the same pool of tasks as curriculum methods but presented in a random ordering?
> 2/ If the non-curriculum methods are trained only on the single target task, what is the role of boosting here, i.e., how is the sum of residuals of different tasks applicable here?
> 3/ “The boosted methods that do not use a curriculum will introduce the additional approximators at the same iteration at which the curricula switches between tasks.” -- this statement is not clear to me. Please clarify.
>
> 1. Exactly, the non-curriculum methods are only trained on the target task.
> 2. Even when only training on a single task, boosting has been shown beneficial to the quality of the $Q$-function approximation in [BOOST]. Hence, the ablations of only using boosting and only using a curriculum serve to show that it is the combination of boosting and a curriculum that yields the best performance.
> 3. When using boosting in a single task, there is still a need to decide at which number of environment interactions additional residuals are added. Since the boosting method is thought of as an ablation to the boosted curriculum, we add the residuals at the same number of environment interactions as in the boosted curriculum - i.e. at every switch in learning task in the curricula.
>
> [BOOST] Tosatto, Samuele, et al. "Boosted fitted q-iteration." International Conference on Machine Learning. PMLR, 2017.
>
> > A question/suggestion/comment regarding the Maze experiment: “It obtains a dense reward based on the distance to the target ...” Isn’t a dense reward already ease the learning process of the RL agent? Of course, here, the size of the two walls will alter the hardness of the environment (by changing the transition dynamics). Still, considering the target task with a sparse reward that obtains a reward only when it reaches the goal would make it a complex problem to solve. Have you tested the sequence of tasks with such sparse rewards?
>
> The outlined experiments in a sparse reward version of the maze or a chain environment are indeed highly interesting. We focused on the dense reward setting in the maze, as it actually introduces a local optima in the RL objective, to which FQI and B-FQI can be seen to converge. This local optima arises since moving around the second wall to reach the goal leads to transient low rewards as the agent needs to temporarily move away from the goal position. Consequently, the dense reward - just like a sparse reward - introduces a challenging exploration problem in the target task. Nonetheless, the sparse reward tasks described by the reviewer are well suited for future work on the topic.

---

> > ### Comment · Reviewer_TFaS · 2021-11-21
> > **Thank you for the response!**
> >
> > I thank the authors for the clarifications and additional discussions on related transfer learning methods in Appendix A.
> >
> > Do you think even the following simplified version of (Brys et al., 2015) would have additional complexities compared to your algorithm (or RPL and PPR):
> > (i) you have a sequence of tasks with increasing difficulty all with sparse rewards.
> > (ii) you solve the first task with a value-based reinforcement learning method. The converged/resulting value function of this task acts as a potential/shaping function to shape the sparse reward of the second task. This shaping operation doesn't bring any additional complexity (Ng et al. 1999).
> > (iii) repeat steps (i) and (ii) in the sequence.
> >
> > [Ng et al. 1999] Ng, Andrew Y., Daishi Harada, and Stuart Russell. "Policy invariance under reward transformations: Theory and application to reward shaping." Icml. Vol. 99. 1999.

---

> > > ### Author Response · Authors · 2021-11-22
> > > **Additional experiment added**
> > >
> > > Thank you! This simplified version of the algorithm by Brys et al. indeed does not share the same complexities as their version. We added an additional empirical evaluation on the LQ problem in the revised appendix (we could move it in the main paper if the reviewer will consider this more appropriate), where we show that the concept of shaping rewards is orthogonal to boosting, and can hence be combined to further improve the performance of boosting and regular learning alike.

---

> > > > ### Comment · Reviewer_TFaS · 2021-11-22
> > > > **Thank you!**
> > > >
> > > > Thank you for the additional experiment! Figure 5 is interesting. In the final draft, please incorporate a discussion on this in the main paper.

---

> > > > > ### Author Response · Authors · 2021-11-24
> > > > > **Thank you!**
> > > > >
> > > > > We agree with the reviewer that the additional experiment shows interesting results, and we thank him/her for proposing it. As suggested, we will incorporate a discussion about it in the main paper for the final draft.

---

### Public Comment · ~Vitaly_Kurin1 · 2021-11-15
**A very relevant piece of work missing**

The paper misses a very relevant piece of work: Silver, Tom, Kelsey Allen, Josh Tenenbaum, and Leslie Kaelbling. "Residual policy learning.", 2018.

---

> ### Author Response · Authors · 2021-11-18
> **Answer to Vitaly Kurin**
>
> We are thankful for this comment as residual policy learning is indeed a related algorithm both conceptually (as it is also based on the idea of learning residuals, however in the form of actions) and from a practical view (as it can be easily incorporated into a curriculum to facilitate transfer without additional effort). Consequently, we evaluated this method in Section 6.3 (please see the answer to reviewer N1P2 for more details).

---

### Decision · Program_Chairs · 2022-01-20

**Decision:**

Accept (Poster)

**Comment:**

The paper proposes a novel curriculum learning method for RL based on the concept of boosting. The proposed method builds on the curriculum value-based RL framework and uses boosting to reuse action-values from previous tasks when solving the current task. The method is analyzed theoretically in terms of approximation accuracy and convergence. Moreover, extensive experiments demonstrate the effectiveness of the method. The reviewers acknowledged the importance of the studied problem setting and generally appreciated the results. I want to thank the authors for their detailed responses that helped in answering some of the reviewers' questions and increased their overall assessment of the paper. At the end of the discussion phase, there was a clear consensus that the paper should be accepted. The reviewers have provided detailed feedback in their reviews, and we strongly encourage the authors to incorporate this feedback when preparing a revised version of the paper.